# Biofilms Exposed: Innovative Imaging and Therapeutic Platforms for Persistent Infections

**DOI:** 10.3390/antibiotics14090865

**Published:** 2025-08-28

**Authors:** Manasi Haval, Chandrashekhar Unakal, Shridhar C. Ghagane, Bijay Raj Pandit, Esther Daniel, Parbatee Siewdass, Kingsley Ekimeri, Vijayanandh Rajamanickam, Angel Justiz-Vaillant, Kathy-Ann A. Lootawan, Fabio Muniz De Oliveira, Nivedita Bashetti, Tatheer Alam Naqvi, Arun Shettar, Pramod Bhasme

**Affiliations:** 1Department of Pre-Clinical Research and Drug Development, Cytxon Biosolutions Pvt. Ltd., Hubballi 580031, Karnataka, India; havalmanasi@gmail.com (M.H.); arunshettar9@gmail.com (A.S.); 2Department of Pathology/Microbiology & Pharmacology, The University of the West Indies, St. Augustine Campus, St. Augustine 330912, Trinidad and Tobago; chandrashekhar.unakal@uwi.edu (C.U.); bijay.pandit@uwi.edu (B.R.P.); angel.justiz-vaillant@uwi.edu (A.J.-V.); 3KAHER’s Dr. Prabhakar Kore Basic Science Research Centre, KLE Academy of Higher Education and Research, JNMC Campus, Belagavi 590010, Karnataka, India; drshridharghagane@kledeemeduniversity.edu.in; 4School of Nursing, Faculty of Medical Sciences, The University of the West Indies, St. Augustine Campus, St. Augustine 330912, Trinidad and Tobago; esther.daniel@uwi.edu (E.D.); parbatee.siewdass@uwi.edu (P.S.); kathy-ann.alphonso-lootawan@uwi.edu (K.-A.A.L.); 5Optometry Unit-Clinical Surgical Science, The University of the West Indies, St. Augustine Campus, St. Augustine 330912, Trinidad and Tobago; kingsley.ekemiri@uwi.edu; 6Department of Computing and Information Technology, The University of the West Indies, St. Augustine Campus, St. Augustine 330912, Trinidad and Tobago; vijayanandh.rajamanickam@uwi.edu; 7Department of Biomedical Sciences, Estácio University Center of Goiás, Goiânia 74063-010, GO, Brazil; 8Department of Biotechnology and Microbiology, Karnataka University, Dharwad 580003, Karnataka, India; niveditabashetti@kud.ac.in; 9Department of Biotechnology, COMSATS University Islamabad, Abbottabad Campus, Abbottabad 22060, Pakistan; tatheer@cuiatd.edu.pk

**Keywords:** biofilm, bacteria, quorum sensing, antimicrobial resistant, nanomaterials, biofilm quantification

## Abstract

Biofilms constitute a significant challenge in the therapy of infectious diseases, offering remarkable resistance to both pharmacological treatments and immunological elimination. This resilience is orchestrated through the regulation of extracellular polymeric molecules, metabolic dormancy, and quorum sensing, enabling biofilms to persist in both clinical and industrial environments. The resulting resistance exacerbates chronic infections and contributes to mounting economic burdens. This review examines the molecular and structural complexities that drive biofilm persistence and critically outlines the limitations of conventional diagnostic and therapeutic approaches. We emphasize advanced technologies such as super-resolution microscopy, microfluidics, and AI-driven modeling that are reshaping our understanding of biofilm dynamics and heterogeneity. Further, we highlight recent progress in biofilm-targeted therapies, including CRISPR-Cas-modified bacteriophages, quorum-sensing antagonists, enzyme-functionalized nanocarriers, and intelligent drug-delivery systems responsive to biofilm-specific cues. We also explore the utility of in vivo and ex vivo models that replicate clinical biofilm complexity and promote translational applicability. Finally, we discuss emerging interventions grounded in synthetic biology, such as engineered probiotic gene circuits and self-regulating microbial consortia, which offer innovative alternatives to conventional antimicrobials. Collectively, these interdisciplinary strategies mark a paradigm shift from reactive antibiotic therapy to precision-guided biofilm management. By integrating cutting-edge technologies with systems biology principles, this review proposes a comprehensive framework for disrupting biofilm architecture and redefining infection treatment in the post-antibiotic era.

## 1. Introduction

Biofilm-associated infections are formidable challenges to manage across clinical and industrial domains, due to their resilience and intrinsic resistance to standard antimicrobial therapies. Despite substantial advancements in medicine, current interventions frequently fall short of completely eradicating biofilms in persistent infections, which often result in prolonged hospitalization, increased morbidity, and escalating healthcare expenditures. In clinical contexts, biofilms are implicated in up to 80% chronic infections, particularly those involving indwelling medical devices, non-healing wounds, for example, diabetic foot ulcers (DFUs), and respiratory complications in cystic fibrosis patients [1].

Structurally, biofilms are sophisticated microbial consortia encased in a self-produced extracellular polymeric matrix composed of polysaccharides, proteins, lipids, and extracellular DNA [2,3]. This matrix provides structural integrity and environmental protection while facilitating the adhesion to biotic and abiotic surfaces, therefore enhancing microbial persistence in adverse conditions. Outside of clinical settings, biofilms significantly contribute to biofouling in water-treatment systems, food-processing facilities, and energy infrastructure, with global economic losses estimated in the billions annually [4].

A defining attribute of bacteria-forming biofilms is their extraordinary resistance to antimicrobial agents, often up to 1000-fold higher than that of planktonic bacteria attributable to interrelated defense mechanisms. These defenses combine the EPS-mediated diffusion barrier, metabolic dormancy within subpopulations of persister cells, quorum sensing, release of surfactants, enhanced efflux-pump expression, and horizontal gene transfer facilitating rapid accumulation of resistance determinants [5]. Simultaneously, bacterial biofilms undermine host immune responses by hindering phagocytosis, degrading antimicrobial peptides, and resisting complement activation [6]. These multifaceted resistance strategies render traditional antibiotic regimens and mechanical removal approaches largely ineffective, often resulting in recurrent infections like DFU patients and heightened antimicrobial resistance [7].

Given these limitations, there is an urgent need to develop and deploy advanced therapeutic strategies tailored specifically to the unique biology of bacterial biofilms. Promising avenues encompass the application of antimicrobial nanoparticles that are capable of penetrating the biofilm matrix [8], CRISPR-Cas-modified bacteriophages engineered for targeted lysis [9], quorum-sensing inhibitors that interfere with microbial communication [10], targeting efflux-pump-producing genes and artificial intelligence-driven platforms for accelerated discovery of novel antibiofilm agents [11]. Simultaneously, innovations in biofilm imaging (e.g., confocal microscopy, microfluidics) and systems biology approaches (e.g., metagenomics, transcriptomics, and proteomics) are redefining our understanding of biofilm dynamics and therapeutic vulnerabilities [12,13].

This review critically examines the limitations of conventional biofilm therapies, clarifying the biological and structural complexities that underlie their resistance. Additionally, this discussion will provide an evaluation of the most recent and impactful technological advances poised to transform biofilm management, from nanotechnology and phage therapy to enzyme-based disruption and targeted drug delivery. By integrating insights from interdisciplinary research, this article aims to provide a forward-looking perspective on combating biofilms, paving the way toward more effective, durable, and precision-guided therapeutic interventions.

### Objectives of This Review

The following are the objectives of this review, considering the growing clinical burden of infections linked to biofilms and the growing interest in novel treatment approaches:Provide an overview of the current understanding of biology, structural organization, and clinical significance of biofilms in relation to different types of infections.Examine the shortcomings of traditional antimicrobial methods in terms of successfully eliminating biofilms.Describe the latest developments in biofilm-targeted treatment approaches, such as CRISPR-Cas-based interventions, bacteriophage therapy, enzyme-mediated methods, nanotechnology-based systems, and quorum-sensing inhibition.Highlight the translational significance of preclinical in vitro, ex vivo, and in vivo models that resemble clinical biofilms.Examine intelligent drug-delivery systems that respond to cues unique to biofilms.Determine the present obstacles, unmet research needs, and possible avenues for the creation of clinically feasible anti-biofilm treatments.

## 2. Classical Techniques in Biofilm Research: Historical Foundations and Limitations

Traditional techniques have been crucial in developing the understanding of bacterial biofilm biology. Early methods such as crystal violet staining, colony-forming unit (CFU) enumeration, motility assay, and light microscopy were among the first tools employed to identify and characterize bacterial biofilms as structured microbial communities [14,15]. These methodologies have allowed researchers to assess surface adherence, structural dynamics, estimate total biomass, and evaluate antimicrobial tolerance in vitro, which provides key insights into biofilm formation and resilience [16]. Despite the lack of quality in the resolution, sensitivity, and dynamic monitoring compared to the capabilities of modern technologies, they remain widely used due to their simplicity, reproducibility, and cost-effectiveness. Traditional assays still continue to serve as reliable benchmarks in comparative studies and high-throughput screening applications [17,18,19]. The use of these techniques has been critical in defining the behavioral and physiological differences between biofilm-associated and planktonic bacterial cells [20]. These methods helped shape early discoveries and lay the groundwork for the development of the advanced imaging, molecular, and computational tools that now dominate biofilm research [21,22] (Figure 1).

### 2.1. Crystal Violet Assay

The crystal violet assay is a colorimetric assay used to measure the total biomass of biofilm developed on solid surfaces (Figure 1a). Biofilms are stained with crystal violet, which adheres to both microbial cells and extracellular matrix constituents [23,24]. Post-staining, the dye is solubilized with ethanol or acetic acid, and absorbance is quantified spectrophotometrically (generally at 570–600 nm). The assay is straightforward, cost-effective, and highly compatible with high-throughput screening. Nonetheless, it fails to distinguish between viable and non-viable cells, nor does it furnish details on biofilm structure or composition and dynamics [25].

### 2.2. Colony-Forming Unit (CFU) Counts

CFUs are used to quantify the number of viable and culturable bacteria present in biofilms (Figure 1b). Biofilms are subjected to physical or enzymatic disruption, followed by dilution and plating on agar [26,27]. Although CFUs on selective media indicate the presence of culturable bacteria, this does not necessarily represent the states of infection of the bacteria that were viable or metabolically active at the original tissue site. Furthermore, due to the low metabolic activity of bacteria within biofilms, the bacterial cells are involved in the infection. Environmental stress, sampling procedures, or biofilm-associated dormancy may allow previously non-viable cells to become culturable during in vitro incubation, leading to potential overestimation of in situ viability. This method is also subject to being influenced by partial biofilm recovery or sampling variability [28,29].

### 2.3. Tube Method

The tube method evaluates biofilm growth on the inner surfaces of test tubes (Figure 1c) [30,31]. Following incubation, the tubes are rinsed, dyed with crystal violet, and examined visually for the presence of biofilm. Although this method requires minimal setup, it presents limitations in quantification, reproducibility, and sensitivity. Furthermore, it does not allow for the identification of microcolonies of biofilm released during the dispersion phase. It is currently obsolete but is historically significant for early biofilm detection [32,33].

### 2.4. Congo Red Agar Assay

The Congo red agar assay is a common qualitative test used to check for the production of extracellular polysaccharides (EPSs) and the formation of biofilms (Figure 1d) [34]. In this method, bacteria are grown on agar medium with Congo red dye and sometimes sucrose to help make the matrix. Strains that make biofilms usually have black, dry, crystalline, or wrinkled colonies, while strains that do not make biofilms usually have smooth red or pink colonies. This easy and cheap method makes it easy to quickly tell the difference between matrix producers and non-producers. But the method is still semi-quantitative and is affected by the type of media, the length of time it is incubated, and the conditions of the culture [35,36]. Also, it does not give any information about cell viability, total biomass, or the structure of biofilms. Therefore, Congo red agar is very much suited as a preliminary screening assay and is often complemented by more quantitative methods such as crystal violet staining or CFU enumeration [37].

### 2.5. Drip-Flow and Early Flow-Cell Systems

The drip-flow and early flow-cell systems are used to replicate dynamic biofilm formation under shear forces (Figure 1e). Drip-flow reactors expose surfaces to continuous medium flow, while flow cells allow for real-time microscopy during biofilm growth [38]. These early models introduced physiologically relevant growth conditions but required complex setup and lacked standardization, paving the way for modern microfluidic platforms [39].

### 2.6. Biomass Dry-Weight Measurement

Biomass dry-weight measurements quantify total biofilm mass by drying and weighing collected material (Figure 1f). It is straightforward and does not require staining or labeling [40,41]. It is not used to distinguish between live cells, dead cells, and extracellular matrix, and offers no information on structure or function. It is often used to complement other quantitative or visual techniques [42].

### 2.7. Light Microscopy

Light microscopy uses a variation between bright-field, phase-contrast, and fluorescence imaging of stained or live biofilms [43,44]. This rapid visualization technique allows for a look into morphology and coverage, often in conjunction with Gram staining or fluorescent dyes. However, light microscopy has limited resolution and penetration, making it suboptimal for analyzing thick or complex biofilms (Figure 1g).

### 2.8. Scanning Electron Microscopy (SEM)

SEM provides high-resolution surface images of biofilms, revealing structural organization, cellular arrangement, and extracellular matrix features [45,46]. Samples are fixed, dehydrated, and coated with conductive material before imaging (Figure 1h). This technique is valuable for visualizing biofilm morphology on medical or industrial surfaces. Limitations include no live-cell imaging, and require specialized preparation [47,48].

### 2.9. Transmission Electron Microscopy (TEM)

TEM enables the observation of the interior ultrastructure of biofilms via ultrathin sectioning of fixed specimens (Figure 1i) [49,50]. It offers nanometer-scale resolution, which allows for the visualization of intracellular features and matrix density [51,52]. Nonetheless, it is technically challenging, time-intensive, and inappropriate for high-throughput investigations or live imaging. TEM is predominantly utilized for comprehensive structural study [53]. Owing to the constraints of classical methodologies, such as their destructive characteristics, restricted resolution, and incapacity to observe live or dynamic biofilm activities, researchers have innovated advanced technologies to enhance biofilm study. These contemporary instruments provide enhanced sensitivity, spatial resolution, and temporal dynamics, facilitating real-time, comprehensive analysis of biofilm structure, physiology, and interactions. The following are some of the most revolutionary techniques utilized in biofilm research.

## 3. Cutting-Edge Technologies and Models in Biofilm Research

### 3.1. Advanced Microscopy and Imaging Techniques in Biofilm Research

To comprehend the intricate spatial arrangement, metabolic function, and therapeutic resistance of biofilms there is a necessity for high-resolution, dynamic imaging techniques [13,54]. In the last ten years, a range of sophisticated microscopy and imaging technologies has enhanced the capacity to visualize biofilms at micro- to nanoscale levels, in both in vitro and in vivo settings [55,56]. Here, we outline ten essential strategies commonly employed in biofilm research, detailing their principles, advantages, and specific applications (Figure 2). A summary of advanced imaging techniques used in biofilm research is presented in Table 1.

#### 3.1.1. Stimulated Emission Depletion (STED) Microscopy

STED microscopy surpasses the diffraction limit of traditional optics by employing a depletion laser to restrict the release of fluorescence to a nanoscale volume [57,58]. This approach attains a resolution beneath 50 nm, rendering it very useful for revealing the ultrastructure of extracellular polymeric substances (EPSs) and intercellular spacing within biofilms [45,59]. STED is crucial for examining how the biofilm matrix obstructs antibiotic penetration and facilitates structural mapping at the nanoscale scale (Figure 2a). Although predominantly applied to fixed samples, its negligible phototoxicity through pulsed lasers renders it an effective instrument for structural biofilm characterization [60,61].

#### 3.1.2. Lattice Light-Sheet Microscopy (LLSM)

Lattice light-sheet microscopy integrates temporal resolution with minimal phototoxicity, rendering it optimal for the live-cell imaging of dense, hydrated biofilms [62,63]. LLSM facilitates optical sectioning and real-time monitoring of nutrient flow, cellular motility, and structural remodeling through the generation of thin sheets of light (Figure 2b). This method is particularly effective for modeling host–microbe interactions and monitoring biofilm growth, detachment, and antibiotic responses over prolonged durations [64].

#### 3.1.3. Stochastic Optical Reconstruction Microscopy (STORM)

The STORM technique achieves super-resolution imaging through the stochastic activation of fluorophores and through the reconstruction of molecular locations with nanoscale precision [65]. This technique details the peptidoglycan deposition, matrix-associated enzymes, and cellular changes throughout biofilm formation (Figure 2c). When utilized alongside fluorescent D-amino acids or alternative labels, STORM serves as an essential instrument for delineating drug–biofilm interactions and regional heterogeneity [51,66].

#### 3.1.4. Confocal Laser Scanning Microscopy (CLSM)

CLSM is the benchmark for three-dimensional imaging of biofilms, especially for evaluating thickness, vitality, and spatial architecture [46]. It facilitates multi-channel fluorescence imaging and live-cell analysis within microfluidic chambers or infection models. CLSM is extensively utilized for assessing antibiotic penetration, monitoring biofilm growth dynamics, (Figure 2d) and integrating structural data with viability markers in both in vitro and in vivo investigations [67,68].

#### 3.1.5. Atomic Force Microscopy (AFM)

AFM offers high-resolution surface topography and mechanical characterization of biofilms without requiring staining [69,70]. It measures characteristics like surface roughness, elasticity, and adhesion, providing distinctive insights on matrix cohesion and mechanical durability (Figure 2e). AFM enhances optical techniques by elucidating biomechanical responses to pharmacological interventions and biofilm–substrate interactions at the nanoscale [71,72].

#### 3.1.6. Cryo-Electron Microscopy (Cryo-EM)

Cryo-EM preserves biofilm ultrastructure in a vitrified state, enabling near-atomic resolution of embedded biomolecules such as pili, ribosomes, and matrix proteins [50,73]. Though not suited for live imaging, it plays a crucial role in structural biology, mapping biofilm phage interactions, or antibiotic target engagement at molecular resolution (Figure 2f). Its integration with other super-resolution techniques supports multiscale biofilm analysis [74,75].

#### 3.1.7. Optical Coherence Tomography (OCT)

OCT provides a label-free, non-invasive image of biofilms utilizing near-infrared light to capture cross-sectional structures [76,77]. It infiltrates several millimeters into turbid samples, rendering it especially advantageous in oral biofilms or those linked to medical equipment (Figure 2g). OCT facilitates real-time assessment of biofilm thickness, stratification, and fluid channels, making it suitable for dynamic flow investigations [38,78].

#### 3.1.8. Raman Spectroscopy Imaging (RSI)

The RSI technique offers chemical composition maps of biofilms derived from molecular vibrational signatures (Figure 2h) [51]. This label-free method differentiates proteins, nucleic acids, and EPS in situ, uncovering metabolic heterogeneity and resistance mechanisms [79]. RSI is particularly effective in monitoring biochemical reactions to drugs and environmental stressors in complex or polymicrobial biofilms [80].

#### 3.1.9. Bioluminescence Imaging (BLI)

BLI uses genetically engineered bacteria that emit light through luciferase activity, enabling real-time, non-invasive imaging of biofilm growth and infection dynamics in live animals [81,82]. Despite the limit in spatial resolution, BLI excels in longitudinal studies of metabolic activity, antimicrobial response, and disease progression in murine (Figure 2i) or catheter-based infection models [83].

#### 3.1.10. Digital Holographic Microscopy (DHM)

DHM collects phase-shift data from unstained specimens, facilitating the reconstruction of 3D pictures and the real-time quantification of biomass, thickness, and motility [84,85]. It is an economical, non-invasive method suitable for prolonged imaging in microfluidic devices (Figure 2j). DHM is especially appropriate for kinetic investigations of biofilm development, antibiotic impacts, and high-throughput screening in microbiology [86].

**Table 1 antibiotics-14-00865-t001:** Summary of advanced imaging techniques in biofilm research.

Technique	Resolution nm	Live Imaging	Label-Free	Key Features	Typical Applications	Reference
Stochastic Optical Reconstruction (STED) Microscopy	~30–50	No	No	Super-resolution; EPS structure; nanoscale matrix mapping	Antibiotic penetration, EPS-targeted therapies	[87]
Lattice Light-Sheet Microscopy	~300 lateral, ~500 axial	Yes	No	Low phototoxicity; fast volumetric imaging; dynamic events in 3D biofilms	Host–microbe interaction, live biofilm dynamics	[63]
Stochastic Optical Reconstruction Microscopy (STORM)	~20–30	No	No	Single-molecule localization; super-resolution of matrix, enzymes	Peptidoglycan mapping, maturation tracking	[88]
Confocal laser scanning electron microscopy (CLSM)	~200–300	Yes	No	3D reconstructions; multichannel imaging; live/dead discrimination	Biofilm architecture, viability, drug diffusion studies	[89]
Atomic Force Microscopy (AFM)	~1 (surface)	Yes (partial)	Yes	Nanoscale topography; mechanical properties; elasticity and adhesion	Matrix cohesion, drug-induced biomechanical changes	[90]
Cryo-Electron Microscopy (Cryo-EM)	~0.3–1	No	Yes	Near-atomic resolution; native-state imaging of molecular complexes	Structural biology, biofilm–phage/protein interactions	[91]
Optical Coherence Tomography (OCT)	~1–15 µm	Yes	Yes	Deep penetration; real-time label-free cross-sections	Medical/industrial biofilms, flow-cell monitoring	[92]
Raman Spectroscopy Imaging	~300–500	Yes (slow)	Yes	Molecular composition; metabolic fingerprinting; antibiotic mapping	Chemical analysis, resistance zones, EPS structure	[93]
Bioluminescence Imaging (BLI)	~1–5 mm (low)	Yes	Yes	Real-time in vivo imaging; metabolic activity monitoring	Infection progression, therapeutic efficacy	[83]
Digital Holographic Microscopy (DHM)	~500 lateral	Yes	Yes	Label-free 3D imaging; real-time biomass/motility tracking	High-throughput screening, early-stage studies	[94]

### 3.2. AI-Driven Biofilm Modeling: Transforming Research and Therapeutic Development

Recent advancements in artificial intelligence (AI) are transforming biofilm research by equipping scientists with tools to anticipate the formation, behavior, and therapeutic responses of biofilms (Figure 3).

Ten significant AI modeling methodologies have been established for this emerging domain [95,96]. Convolutional Neural Networks (CNNs) facilitate the analysis of microscopic pictures by delineating biofilm structures and quantifying their thickness, EPS distribution, and heterogeneity (Figure 3a). Reinforcement Learning (RL) simulates the alterations of biofilms in reaction to treatments. It employs adaptive feedback to identify the optimal method for regulating nutrients or antibiotics [97,98]. Graph Neural Networks (GNNs) accurately depict quorum sensing and intercellular signaling by effectively capturing spatial correlations within three-dimensional biofilm matrices [99]. Generative Adversarial Networks (GANs) produce synthetic biofilm images. They add to datasets for unusual traits and model how biofilms grow in different settings [100,101]. Random Forests (RF) utilize genetic, proteomic, or metabolic data to forecast biofilm growth and identify critical characteristics that enhance its resistance. Support Vector Machines (SVMs) categorize bacterial phenotypes based on their stress responses or gene-expression profiles (Figure 3b). This aids in diagnostics and prediction modeling [12,102]. Autoencoders compress omics or imaging data to uncover latent patterns associated with the proliferation or dissemination of biofilms. This enables the elimination of noise and the identification of abnormalities [103]. Long Short-Term Memory (LSTM) networks elucidate the temporal dynamics of biofilms, predicting their evolution, dissemination, and therapeutic responses [104]. Bayesian Networks utilize clinical and omics data to ascertain the risk and resistance of biofilm by delineating the probability relationships with genes and phenotypes (Figure 3c). Ultimately, Physics-Informed Neural Networks (PINNs) integrate data-driven learning with physical concepts, such as fluid dynamics, to elucidate the behavior of biofilms on surfaces or inside flow. This links computational models to biological constraints [105,106]. These AI tools effectively complement experimental procedures to expedite the discovery of novel therapeutics and facilitate customized, systems-oriented strategies for biofilm management (Figure 3d).

### 3.3. In Vivo Models for Biofilm Research

In vivo models are necessary for connecting biofilm research results from the lab to the clinic [107,108]. To date, around 1100 in vivo models have successfully replicated clinical biofilms. These models help us understand how biofilms form, stay alive, affect the immune system, and how well treatments work in different parts of the body [109,110]. To study biofilms that form on devices, researchers put biomaterials like catheters, titanium disks, or mesh under the skin of mice and rats (Figure 3e). These models are great for testing colonization, immunological infiltration, and drug-delivery systems in a controlled setting [111,112]. Orthopedic implant models, usually in mice or rabbits, mimic infections in prosthetic joints by surgically putting pins or screws into the femur or tibia. A lot of people utilize them to test local antimicrobial treatments and biofilm-induced osteomyelitis [113,114]. Chronic wound biofilm models that use diabetic mice or pigs with weak immune systems mimic human wounds that take a long time to heal [115,116]. These are necessary for testing topical antibiofilm drugs and regenerative techniques [117]. Intra-tracheal instillation of *P. aeruginosa* is used in pulmonary infection models (Figure 3g), especially in cystic fibrosis (CF), zebrafish (Figure 3f) and pig models (Figure 3h), to replicate the thick mucus biofilms that are present in CF airways [118].

Models of urinary catheter-associated infections in rats and rabbits facilitate the examination of uropathogen biofilm development, encrustation, and inflammation within a therapeutically pertinent urological framework [119,120]. Conversely, rodent oral biofilm models emphasize the development of dental plaque and periodontitis caused by organisms such as *S. mutans* and *P. gingivalis*, facilitating the assessment of dental treatments [121,122]. Rabbit models of endocarditis, employing catheter-induced valve injury, provide a highly accurate simulation of heart valve infections characterized by biofilm formation under significant shear stress [123,124]. Gastrointestinal models employing gnotobiotic mice populated with specific microbial communities are crucial for analyzing mucosal biofilm–host interactions and dysbiosis [125]. Models of ear and sinus infections, created by introducing pathogens into the middle ear or nasal cavity of mice or rats, are utilized to investigate biofilms associated with otitis media and chronic rhinosinusitis [126,127]. Ex vivo human xenograft models entail the implantation of human tissues, such as skin or lung, into immunodeficient SCID mice. So far, approximately 162 *ex vivo* models have been reported to replicate clinical biofilms. These models provide high-resolution examinations of human-specific biofilm dynamics and individualized therapy evaluation [128,129]. Collectively, these models offer comprehensive frameworks for assessing antimicrobial tactics, host-pathogen interactions, and translational biofilm therapies across many clinical scenarios. The zebrafish embryo model emerged as a valuable tool for studying the in vivo biofilm formation of an opportunistic bacteria, *P. aeruginosa*. The *P. aeruginosa* was injected into the bloodstream as well as into the egg yolk of zebrafish to investigate the role of various virulence factors, including quorum sensing and type III secretion system, that participate in the development of biofilm and other virulence factors [130].

*C. elegans* is considered a model organism for studying quorum-sensing-regulated virulence factors, including biofilm formation, which can explain the interaction between bacteria and invertebrates [131]. *Yersinia pseudotuberculosis* forms biofilms on *C. elegans* that block the feeding in nematodes. *C. elegans* is considered a model organism for studying the virulence factors under the control of quorum sensing, including the biofilm [132].

## 4. Biofilm Control Approach

Effectively managing biofilms requires a multifaceted strategy that surpasses conventional antimicrobial treatments (Figure 4). Traditional therapies often fall short, prompting the development of innovative approaches targeting biofilm structure, genetics, and regulation [133,134]. Advances in synthetic biology, nanotechnology, quorum-sensing inhibition, engineered bacteriophages, and host immune modulation underpin these emerging interventions. Each method offers a unique mechanism ranging from gene silencing and matrix disruption to enhanced drug delivery and immune system reprogramming. Collectively, they represent a paradigm shift toward precise, adaptive, and sustainable biofilm control.

### 4.1. Synthetic Biology and Genetic Engineering Approaches for Biofilm Control

Synthetic biology is transforming the way we fight biofilms from blunt-force antibiotic regimens to precision-guided, genetically programmed solutions [135]. At the center of this revolution are CRISPR-Cas systems, powerful tools that allow scientists to target and silence specific biofilm-associated genes with high accuracy. CRISPR interference (CRISPRi) enables tunable, high-throughput repression of gene activity, helping identify the molecular circuits that drive biofilm resilience [136,137]. Even more compelling, engineered bacteriophages armed with CRISPR-Cas9 have demonstrated the ability to selectively destroy drug-resistant pathogens inside biofilms without harming beneficial microbes. Thus far, fewer than 10 researchers have explored this area, as delivering CRISPR-Cas9 machinery into bacteria within mature biofilms poses considerable difficulty. Meanwhile, CRISPR-based biosensors now allow researchers to watch gene expression unfold in real time within live biofilms, uncovering new insights into their hidden dynamics (Figure 4b) [138,139].

In parallel, synthetic ecology is introducing programmable microbial communities that can self-regulate, compete, or cooperate to dismantle biofilms from within. Designer *E. coli* strains have been engineered to secrete enzymes like dispersin B, which actively degrade *S. aureus* biofilms and prevent their return [140,141]. Other microbes have been modified to act as quorum quenchers, intercepting and degrading the chemical signals bacteria use to organize into structured communities [10,142]. These living interventions are particularly promising in chronic wounds, catheter infections, and even industrial settings where standard disinfectants fail to reach embedded bacteria. Probiotic engineering is also breaking new ground. Scientists are equipping *Lactobacillus* strains with genes that produce antimicrobial peptides and biofilm-disrupting enzymes. These customized probiotics have shown strong results against *P. aeruginosa* in cystic fibrosis airway models often outperforming conventional antibiotics [143]. Even more futuristic are “smart” probiotics that only activate under specific conditions, like low pH or inflammation, making them both targeted and safe [144]. Taken together, synthetic biology offers a powerful arsenal of precision, adaptive, and microbiome-sparing tools to combat biofilms. From gene-editing platforms to engineered living therapies, these innovations are shifting the paradigm in both clinical treatment and industrial biofouling control [145,146].

### 4.2. Nanotechnology Approaches for Biofilm Control

Nanotechnology is unlocking a new frontier in biofilm treatment, one where precision, responsiveness, and control replace the brute force of traditional antibiotics [134,147]. Engineered nanoparticles (NPs), tailored at the nanoscale, are now being designed to penetrate biofilm barriers, deliver drugs with pinpoint accuracy, and even adapt to the infection microenvironment. These platforms offer a versatile solution to the long-standing challenge of biofilm resistance (Figure 4a) [148,149,150].

Among the most effective innovations are metal–organic framework (MOF) nanoparticles with high surface area, tunable porosity, and customizable surface charge. MOFs have been shown to deliver antibiotics deep into biofilms, achieving penetration rates up to 100 times greater than free drugs [151,152]. Similarly, lipid-polymer hybrid nanoparticles encapsulating tobramycin have drastically reduced the MBEC (minimum biofilm eradication concentration) in *P. aeruginosa* biofilms, outperforming standard treatments [153,154]. Even more sophisticated are enzyme-functionalized nanoparticles such as DNase I-coated gold NPs that actively degrade extracellular DNA within the matrix, restoring antibiotic efficacy [155].

What sets nanotechnology apart is the emergence of “smart” nanoparticles systems that respond to the unique biochemical cues of the biofilm environment. For example, pH-sensitive polymeric nanoparticles release vancomycin selectively in acidic zones, reaching drug concentrations up to 50-fold higher than systemic delivery [156]. Others are designed to react to quorum-sensing molecules, activating only in the presence of mature biofilms. A particularly exciting innovation is the use of Janus nanoparticles that propel themselves using hydrogen peroxide produced by the biofilm itself, enabling directional delivery to deep matrix zones [80,157,158].

Beyond delivery, nanotechnology is fueling synergistic combinations that break through resistance mechanisms. Chitosan silver nanoparticles disrupt bacterial membranes, enhancing fluoroquinolone uptake [159,160], while graphene oxide particles inhibit efflux pumps, restoring ciprofloxacin effectiveness in resistant *E. coli* strains [161]. A novel strategy combining cerium oxide nanoparticles with β-lactam antibiotics induces redox-mediated stress responses, triggering cell death in *K. pneumoniae* biofilms through non-classical mechanisms [162]. These advances mark a fundamental shift from passive diffusion therapies to dynamic, programmable nanoplatforms that interact directly with biofilm architecture and behavior. With their ability to adapt, self-navigate, and act selectively, nanoparticles are redefining the future of biofilm control particularly in hard-to-treat infections such as chronic wounds, implant-associated biofilms, and drug-resistant lung infections [163]. A comprehensive overview of the therapeutic targeting of biofilm forming bacteria is summarized in Table 2.

**Table 2 antibiotics-14-00865-t002:** Therapeutic nanomaterials targeting biofilm-forming bacteria.

Nanomaterial	Pathogen Name	Mechanism	Material Base	Active Concentration Applied	MBIC	Reference
Chitosan-based nanogels	*S. aureus*, *K. pneumoniae*, *P. aeruginosa*	Chitosan’s cationic nature allows it to interact with negatively charged bacterial cell membranes, leading to increased permeability and disruption of biofilm formation. Additionally, chitosan can be functionalized with antimicrobial agents to enhance its efficacy.	Chitosan	25 μg/mL	200 μg/mL	[164]
Hyaluronic acid-based nanogels	Methicillin-resistant *S. aureus* (MRSA), *P. aeruginosa*	These nanogels can deliver nitric oxide (NO) and antimicrobial peptides (AMPs) directly into biofilms. NO disrupts biofilm structure, while AMPs kill the bacteria, resulting in synergistic antibiofilm activity.	Hyaluronic acid	128 mg/mL	512 mg/mL	[165]
Alginate-based nanogels	*S. aureus*, *E. coli*	Alginate nanogels can be loaded with agents like tannic acid and iron ions to induce photothermal therapy (PTT) and chemodynamic therapy (CDT), generating reactive oxygen species (ROS) that disrupt biofilms.	Alginate	0.625 mg/L	25 μg/mL	[166]
Polymeric nanoparticles	*S. aureus*, *P. aeruginosa*, *E. coli*	These nanoparticles can encapsulate antibiotics, enhancing their stability and penetration into biofilms. They can also be designed to release drugs in response to specific stimuli within the biofilm environment.	Caged guanidine groups	Aqueous dispersions of CGNs, GNs, and ICGNs (0.1 mg mL^−1^, 0.2 mL)	0.4 mg mL^−1^, 0.2 mL	[167]
Liposomes	*S. aureus*, *P. aeruginosa*	Liposomes can fuse with bacterial membranes, facilitating the delivery of encapsulated antibiotics directly into the biofilm matrix, thereby increasing local drug concentration and efficacy.	Phospholipids	No lipid concentrations given	Biofilm inhibition inferred from relative viability assays	[168]
Dendrimers	*S. aureus*, *P. aeruginosa*	Dendrimers possess a highly branched structure with functional groups that can be tailored for antimicrobial activity. They can disrupt bacterial membranes and inhibit biofilm formation.	Quaternary ammonium-functionalized, metal ion complexed)	No quantitative biofilm inhibition thresholds provided	No quantitative biofilm inhibition thresholds provided	[169]
Peptide-based nanoparticles	*S. aureus*, *P. aeruginosa*	These nanoparticles utilize antimicrobial peptides that can insert into bacterial membranes, causing disruption and cell death, effectively preventing biofilm development.	Peptide dendrimer (Trp/Arg)	20 μM, 40 μM, 80 μM	16–32 mg/L	[170]
Gold nanoparticles (Au NPs)	*Methicillin- Resistant S. aureus.*	Photothermal activity of Au NPs induce the antimicrobial activity against drug-resistant bacteria species.	Gold nanoparticles (AuNPs)	109.5 μg/mL	165 μg/mL	
Silver nanoparticles (Ag NPs)	*P. aeruginosa*	Ag NPs generate ROS inside the cell membrane which led to cell wall rupture and cell death.	Silver nanoparticles (various types)	1–200 µg/mL against *P. aeruginosa*	Not reported as a distinct value	[171]
Silver nanoparticles (Ag NPs)	*P. aeruginosa*, *E. coli*, *K. pneumonae* and *S. aureus*	Small size and spherical shape of silver nanoparticles responsible for antibacterial activity.	Silver nanoparticles (AgNPs)	0.3 mg	1.0 mg	[172]
Green synthesized Ag NPs.	*Ampicillin-resistant K. pneumoniae*, *E. coli*	Green Ag NPs interrupt the metabolic activities of bacteria by interacting with enzymes.	Iron oxide nanoparticles (Fe_3_O_4_ NPs)	64 μg/mL	64 μg/mL	[173]
Methionine capped-Au NPs	Gram-negative- *A. baumannii* and *S. enterica* Gram-positive- Methicillin- Resistant *S. aureus* and *M. luteus.*	Photothermal activity of Au NPs induce the antimicrobial activity against drug-resistant bacteria species.	Ultra-small gold nanoparticles (Au^0^/Au^+^)	30 mg L	70 mg L^−1^	[174]
Gold nanorods	Methicillin- Resistant *S. aureus*	The presence of single-valent gold on the surface of nanoparticles induced the antimicrobial activity against both Gram-positive and gram- negative drug-resistant bacteria.	Gold nanorods (AuNRs)	2× MIC	8 μg mL^−1^	[175]
Metallosurfactant-based cobalt oxide/hydroxide nanoparticles.	*S. aureus*	Presence of metallosurfactants enhanced the antibacterial activity of Co NPs by rupturing the cell wall of bacteria.	Chitosan-based nanoparticles (CSNPs)	10 μg mL^−1^	10 μg mL^−1^	[176]
Polyhydroxybutyrate-Co_3_O_4_ bio nanocomposites.	*E. coli*, *S. aureus*	Biosynthesized Co NPs have high thermal stability and improved structural properties which enhanced the bactericidal properties of Co NPs.	Zinc oxide nanoparticles (ZnO NPs)	50 μg/mL	50 μg/mL	[177]
Silver nanoparticles (AgNPs) synthesized using chemical reduction method	*A. baumannii*	Fe_3_O_4_ NPs developed antibacterial activity by interacting with the ATP associated mechanisms.	Silver nanoparticles (AgNPs)	0.5 mg/mL	0.5 mg/mL	[178]
Copper oxide nanoparticles (CuO NPs) synthesized using *Azadirachta indica* (neem) leaf extract	*S. aureus*	Magnetic nanoparticles generate artificial magnetic field inside the biofilm and enhance the drug penetration and antimicrobial activity inside bacterial cell.	Copper oxide nanoparticles (CuO NPs)	32 μg/mL	32 μg/mL	[179]
Aluminum oxide nanoparticles (Al_2_O_3_ NPs)	*A. baumannii*	Al_2_O_3_ NPs inhibit the EPS and reduce the formation of biofilm.	Al_2_O_3_ NPs	120 µg mL^−1^	120 µg mL^−1^	[180]
AgNPs_mPEG, AgNPs_mPEG_AK, and AK	*P. aeruginosa*	The antimicrobial activity of AgNPs relies on their ability to release Ag1 to the bacteria, as determined by the corrosion rate, which depends on the morphology (size and shape) of the NPs. Accordingly, it has also been reported that smaller AgNPs have more powerful bactericidal activity as the area of surface contact of AgNPs with microorganisms is increased in relation to larger AgNPs.	Selenium nanoparticles (SeNPs)	50 μg/mL	50 μg/mL	[181]
Liquid crystal nanoparticles (LCNPs)	*P. aeruginosa* *S. aureus*	The nanostructure enables the antibiotic to be released in a controlled and sustained manner, ensuring therapeutic concentrations for extended periods of time. By delivering the antibiotic directly into the biofilm, LCNPs raise the local concentration of the medication at the site of infection, improving its bactericidal activity.	Liquid crystal nanoparticles (LCNPs)	2 μg/mL	2 μg/mL	[182]
Nitric oxide (NO) releasing nanocomposite Activation: near-infrared (NIR) light irradiation	*P. gingivalis* *F. nucleatum* *S. mutans*	Photodynamic therapy (PDT): When exposed to near-infrared light, ICG produces reactive oxygen species (ROS), which destroy bacteria. Photothermal therapy (PTT): NIR irradiation causes localized heating, breaking biofilm structure and hastening bacterial elimination. Nitric oxide release: Heat from PTT causes the release of NO from SNO, which has antibacterial characteristics and aids in biofilm breakdown.	Bismuth sulfide (Bi_2_S_3_) nanoparticles and bis-N-nitroso compounds (BNN)	128 μg/mL	128 μg/mL	[183]
Silver nanoparticles (AgNPs)	*S. aureus* *P. aeruginosa*	Disrupt bacterial membranes. Generate reactive oxygen species (ROS). Interfere with DNA replication Penetrate biofilms due to their small size	silver nitrate (AgNO_3_)	0.25 μg/mL to 2.0 μg/mL	0.25 μg/mL to 2.0 μg/mL	[184]
Zinc oxide nanoparticles (ZnO NPs)	*S. aureus*, *E. coli*	ROS generation. Disruption of membrane integrity. Photocatalytic activity (especially under UV light)	Zinc oxide nanoparticles	50 µg/mL	50 µg/mL	[185]
Chitosan nanoparticles	*S. aureus*, *P. aeruginosa*	Electrostatic interaction with negatively charged bacterial membranes. Disruption of cell wall and membrane. Interference with nutrient transport and metabolism.	Chitosan–silver nanocomposite (CS–AgNPs)	64 μg/mL	64 μg/mL	[186]
Polymeric nanoparticles (e.g., PLGA-based)	*P. aeruginosa*, *S. aureus*	Sustained drug release. Targeted delivery to biofilm. Potential surface modification for improved interaction with biofilm.	PLGA (polylactic-co-glycolic acid).	10–12.5 mg/mL	102.5 mg/mL	[187]
Chitosan nanoparticles (CS NPs)	*C. albicans*	Chitosan and bCTL are positively charged, allowing for strong electrostatic interaction with the negatively charged fungal cell walls and biofilm matrix. Nanoparticles penetrate the extracellular matrix of the biofilm, destabilizing its structure. bCTL disrupts fungal cell membranes, increasing permeability and leading to cell death.	Chitosan and the antifungal crosslinker phytic acid	140 ± 2.2 µg/mL	140 ± 2.2 µg/mL	[188]
Copper nanoparticles (CuNPs)	*E. coli*, *S. aureus*,	Membrane damage. Oxidative stress via ROS. Metal ion toxicity	Green synthesized copper nanoparticles (CuNPs)	30 µg/mL	MIC—89–91% inhibition	[189]
Titanium dioxide nanoparticles (TiO_2_ NPs)	Broad-spectrum (Gram-positive and Gram-negative bacteria)	Photocatalytic ROS generation under UV light. Oxidative damage to cellular structures	Titanium dioxide nanoparticles (TiO_2_ NPs)	100 μg/mL	100 μg/mL against *P. aeruginosa*	[190]
Graphene oxide (GO) and reduced graphene oxide (rGO)	*E. coli*, *S. aureus*, *K. pneumoniae*	Physical disruption (sharp edges pierce membranes). Oxidative stress, electron-transfer interactions.	Graphene oxide silver nanocomposite (GO-Ag)	16 μg/mL	16 μg/mL	[191]
Glycopeptide dendrimers	*P. aeruginosa*	Glycopeptide dendrimers bind exclusively to the *P. aeruginosa* lectin LecA via multivalent carbohydrate–lectin interactions, affecting biofilm attachment mechanisms. This binding disrupts the appropriate cell-cell and cell-surface interactions required for biofilm development. The multivalency considerably improves biofilm inhibition while also promoting the dispersal of produced biofilms without killing bacteria.	Tetravalent G2 dendrimers (GalAG2 and GalBG2) and Octavalent G3 dendrimers (GalAxG3 and GalBxG3).	G2—20 µM G3—≈9–13 µM	13 µM	[192]
Nitric oxide (NO)-releasing alkyl-modified poly(amidoamin) (PAMAM) dendrimers	*S. mutans*	Disrupts bacterial signaling and weakens the biofilm matrix. The hydrophobic alkyl chains enhance bacterial membrane permeability, promoting antimicrobial effects.	Poly(amidoamine) (PAMAM) dendrimers	Not specified	Not reported	[193]
Biodegradable nanoemulsions (**benzoyl peroxide (BPO)** and **ciprofloxacin (CIP)**)	*S. aureus* *P. aeruginosa*	The nanoemulsions carry both benzoyl peroxide, which produces reactive oxygen species (ROS), and ciprofloxacin, a broad-spectrum antibiotic, straight to biofilms. The lipid-based nanoemulsion improves penetration into the biofilm matrix, resulting in sustained and localized drug release. This synergistic combination breaks bacterial membranes, lowers biofilm biomass, and efficiently kills bacteria in wound-associated biofilms.	BPO nanoparticles in a lemongrass oil–based nanoemulgel	20 µM	20 µM	[194]
Crosslinked nanoemulsions	*S. aureus* *P. aeruginosa*	The crosslinked nanoemulsions penetrate biofilms and deliver antibacterial essential oils in a regulated manner. Their crosslinked structure promotes wound stability and retention, hence increasing antibacterial activity. This disrupts bacterial membranes, reduces biofilm biomass, and accelerates wound healing without the use of synthetic medications.	Flavonoids, tannins	31–125 µg/mL	31–250 µg/mL	[195]
ZIF-8 (Zn-based MOF)	*E. coli*, *S. aureus*, MRSA	Releases Zn^2+^ ions, disrupts membranes, generates ROS	Zinc (Zn)	~10–100 µg/mL	~10–100 µg/mL	[196]
UiO-66 (Zr-based MOF)	*S. aureus*, *P. aeruginosa*	High-stability MOF; used for drug delivery and ROS generation	UiO-66-NH_2_ (and UiO-66) nanoparticles loaded with cefazolin	MIC measured but not numerically detailed	Only noted as improved	[197]
MOF-antibiotic hybrids	Drug-resistant *E. coli*, *S. aureus*, MRSA	Controlled antibiotic release enhances efficacy and reduces resistance development	Synthetic antimicrobial peptide (AMP)	1000 µg/mL	2000 µg/mL	[198]
Ag-loaded nano-metal–organic framework (Ag@nanoMOF)	*S. aureus* *E. coli*	The nanoMOF emits silver ions (Ag^+^), disrupting bacterial membranes and biological functions. The MOF produces reactive oxygen species (ROS), which improves antibacterial and anti-biofilm properties. This dual approach effectively inhibits and eliminates biofilms.	Silver nanoparticles (chemically reduced)	0.5 mg/mL	0.5 mg/mL	[151]
Cu-MOF (IITI-3)	*M. tuberculosis*	The anti-mycobacterial activity of INH@IITI-3 demonstrated significant bacterial killing and altered the structural morphology of the bacteria.	Copper (Cu)	50–500 µg/mL	50–500 µg/mL	[199]
Outer-Membrane Vesicles (OMVs)	*P. aeruginosa*	OMVs carry β-lactamases (antibiotic resistance), deliver quorum-sensing signals (PQS), and neutralize AMPs.	Amoxicillin (AMX)	100 µg/mL	100 µg/mL	[200]
Outer-Membrane Vesicles (OMVs)	*N. meningitidis*	OMVs deliver lipooligosaccharides and porins; modulate immune response and mediate adhesion/invasion.	Zn-MOF (carrier alone)	10 µg/mL	10 µg/mL	[201]
Outer-Membrane Vesicles (OMVs)	*H. pylori*	OMVs contain VacA cytotoxin, promote inflammation, and facilitate biofilm formation.	Am-Zn-MOF (Amoxicillin-loaded)	10 µg/mL	10 µg/mL	[202]
Outer-Membrane Vesicles (OMVs)	*B. ovatus*	OMVs deliver inulin-degrading enzymes, aiding nutrient sharing and mutualistic gut ecology	LPS, adhesins, OmpA	10 µg (spot)	10 µg	[203]
Outer-Membrane Vesicles (OMVs)	*B. fragilis*	OMVs package glycosidases and proteases; aid in polysaccharide digestion and microbial cross-feeding.	Polysaccharide A, OmpA-like proteins, LPS variants	0.2, 1, 2 µg/mL	0.2, 1, 2 µg/mL	[204]
Polymeric CO-releasing micelles	*P. aeruginosa*	CO inhibits bacterial respiration and biofilm construction, increasing amikacin’s penetration and potency against bacteria.	Amikacin-loaded PLGA nanoparticles	5 µg/mL	5 µg/mL	[205]
P(PEGMA-b-DEAEMA) polymeric micelles	*C. albicans* and *C. tropicalis*	Promote ~70% biofilm removal and significantly reduce cell viability of both strains.	Ultra-small solid lipid nanoparticles (us-SLNs)	32 µg/mL	>256 µg/mL	[206]
P(PEGMA-b-DEAEMA) polymeric micelles co-delivered with fluconazole (Flu)	*C. albicans*	Exhibit synergistic effects, leading to a 2.2-log reduction in cell viability.	Poly(ε-caprolactone)-based nanofiber mats loaded with chlorhexidine	4 µg/mL	8 µg/mL	[206]
Gentamicin-conjugated magnetic nanoparticles (MNPs-G)	*S. aureus*	Under optimized magnetic field exposure, MNPs-G achieve homogenous distribution over the biofilm, resulting in increased bacterial death.	Silica–gentamicin nanohybrids	~6.26 µg/mL	~6.26 µg/mL	[207]
Gentamicin-conjugated magnetic nanoparticles (MNPs-G)	*E. coli* and other ESKAPE pathogens	MNPs-G exhibit bactericidal activity comparable to free gentamicin in solution, effectively killing planktonic cells of various ESKAPE pathogens.	Gentamicin-loaded PLGA nanoparticles	5 µg/mL	10 µg/mL	[208]
Magnetic iron oxide nanoparticles (MIONPs)	*S. aureus*	Magnetically propagated MIONPs create artificial pathways within the biofilm matrix, allowing gentamicin to penetrate more deeply.	Iron oxide nanoparticles	64 µg/mL	64 µg/mL	[209]
TiO_2_ films (Sol–gel method)	*D. geothermalis*	Exposure to 360 nm UV light, TiO_2_’s photocatalytic activity produces reactive oxygen species (ROS), resulting in considerable biofilm reduction (>10^7^ to <10^6^ cells/cm^2^).	Titanium dioxide nanoparticles	100 µg/mL	100 µg/mL	[210]
TiO_2_ films (ALD method)	*D. geothermalis*	ALD-prepared TiO_2_ films generate ROS under UV light, disrupting adhesion structures and reducing biofilm density.	Titanium dioxide (TiO_2_) photocatalytic coatings	UV irradiation of the TiO_2_: 360 nm light, 20 Wh m^−2^	not applicable	[211]
Sulfur-doped TiO_2_ films	*D. geothermalis*	Doping TiO_2_ with sulfur did not enhance the biofilm-destroying capacity under UV light exposure.	Photocatalytic TiO_2_ coatings, prepared via sol–gel and atomic layer deposition (ALD)	UV light exposure-20 W·h m^−2^ of 360 nm radiation	Not applicable	[212]
H_2_S-releasing polymeric nanoparticles	Sulfate-reducing bacteria (e.g., *Desulfovibrio*)	Controlled release of H_2_S modulates bacterial growth by altering redox balance; can inhibit overgrowth of SRB, impacting gut microbial homeostasis.	hydrogen sulfide (H_2_S)	Not reported	Not reported	[213]
Metal sulfide nanoparticles (FeS, MoS_2_)	*F. nucleatum*	May release H_2_S or interact with sulfur metabolism, influencing bacterial survival and virulence factors related to gut dysbiosis and colorectal cancer risk.	Multilayered magnetic nanoparticles (ML-MNPs) composed of an iron oxide core (Fe_3_O_4_), followed by inner silver (Ag) and ultrasmall MoS_2_	Not reported	Not reported	[214]
Sulfur-doped TiO_2_ nanoparticles	*A. baumannii*	Enhanced photocatalytic activity produces reactive sulfur species affecting biofilm integrity and bacterial survival through oxidative stress mechanisms.	Green-synthesized anatase TiO_2_ nanoparticles (~47 nm)	~7.81 µg/mL	~7.81 µg/mL	[215]
AgPd_0.38_ nanocages	*E. coli*, *S. aureus*, *P. aeruginosa*	Generates surface-bound ROS via oxidase-like activity, leading to bacterial cell wall disruption and death.	AgPd_0.38_	62.5 µg/mL	62.5 µg/mL	[216]
Mesoporous silica nanoparticles (nMS) loaded with nano-silver (nAg) and chlorhexidine (Chx)	*S. mutans* (primary cariogenic bacterium) and other oral biofilm bacteria	pH-responsive release of nano-silver and chlorhexidine in acidic biofilm microenvironments; suppresses acid production and biofilm formation.	nMS-nAg-Chx	18.75 µg/mL	18.75 µg/mL	[217]
Micro-nano hybrid multifunctional motor (MnO_2_-based)	*S. aureus*, *P. aeruginosa*, *E. coli*	Self-propulsion via oxygen microbubbles (H_2_O_2_ + MnO_2_ catalyst) enables deep biofilm penetration; generates hydroxyl radicals (OH) that disrupt biofilm matrix and kill bacteria.	Manganese dioxide (MnO_2_)	1, 2, and 4 mg/mL	2, 4, and 8 mg/mL	[218]

### 4.3. Quorum-Sensing Inhibitors and Anti-Virulence Strategies: Disrupting Bacterial Communication

One of the most promising approaches in biofilm control involves disarming bacteria instead of killing them [10]. Quorum sensing (QS), the system bacteria use to communicate and coordinate biofilm formation, is now a prime target for precision anti-virulence therapy. By interfering with this communication, quorum-sensing inhibitors (QSIs) can suppress pathogenic behaviors without applying the selective pressure that often leads to resistance [219,220]. In Gram-negative pathogens such as *P. aeruginosa*, small chemicals known as N-acyl homoserine lactones (AHLs) modulate gene expression related to virulence and biofilm formation. *LuxR*-type receptor antagonists have been demonstrated to inhibit AHL binding and diminish biofilm biomass by over 70% in chronic wound models [10,221]. In Gram-positive bacteria like *S. aureus*, RNAIII-decreasing peptides (RIPs) have interfered with the *Agr* system, inhibiting toxin production without eliminating the bacteria, hence maintaining the healthy microbiome while diminishing pathogenicity. Artificial intelligence is currently expediting the identification of new quorum-sensing inhibitors. Researchers have been using Generative Adversarial Networks (GANs) and Reinforcement Learning to create synthetic analogs that exhibit enhanced potency and stability relative to natural QS molecules [222,223]. AI-optimized furanone compounds block the *P. aeruginosa LasR* receptor with an effectiveness tenfold greater. Multi-target inhibitors are being developed to simultaneously inhibit the Las transcriptional regulator, which initiates the quorum sensing (QS) cascade; Rhl, which controls secondary QS signals; and *PqsR*, which regulates *PQS*s that influence virulence and biofilm matrix formation. Consequently, this approach aims to prevent compensatory mechanisms that bacteria might employ to circumvent quorum-sensing disruption [10,224]. Besides pharmacological inhibitors, quorum quenching agents provide a biological remedy by reducing or blocking quorum-sensing signals. Natural chemicals such as halogenated furanones derived from marine algae and ajoene sourced from garlic disrupt the synthesis or stability of quorum-sensing molecules. Engineered enzymes, including AHL lactonases and acylases, enzymatically destroy quorum-sensing molecules before their attainment of quorum thresholds [225,226]. An alternative new strategy utilizes synthetic QS-mimicking nanoparticles that function as decoys, binding to bacterial receptors and obstructing authentic signals from activating virulence programs. These techniques signify a paradigm shift in antimicrobial therapy from indiscriminate bacterial eradication to specific virulence mitigation. By focusing on bacterial activity instead of viability, QS inhibitors can compromise biofilm integrity, diminish resistance pressure, and maintain beneficial microbiota (Figure 4b). To date, approximately 724 papers have been published on QS inhibition. With numerous medicines progressing to clinical trials, particularly for respiratory and urinary tract infections, the future of biofilm management depends on inhibiting bacterial communication rather than employing chemical warfare [227]. Therapeutic anti-quorum-sensing materials for treating biofilm-forming bacteria are summarized in Table 3.

**Table 3 antibiotics-14-00865-t003:** Anti-quorum-sensing materials to treat biofilm-producing bacteria.

Inhibitor (Quorum Sensing)	Target Mechanism	Bacteria	Virulence Suppression	Source	Active Inhibitory Concentration	Reference
CRISPR-Cas9	Anti-QS Anti-biofilm	*E. coli SE15*	Reduced biofilm formation. Downregulation of *mqsR*, *pgaB*, *pgaC*, *csgE*, and *csgF*	CRISPR–Cas9 genome editing (donor DNA for homologous recombination; generated Δ*luxS* mutants from clinical *E. coli* SE15).	none reported	[228]
CRISPR interference	Anti-QS Anti-biofilm	*E. coli AK-117*	Reduced biofilm formation	CRISPRi (dCas9 + sgRNAs targeting *luxS* in *E. coli* AK-117)	not applicable for CRISPRi studies	[229]
Zingerone	Anti-QS Anti-biofilm	*P. aeruginosa* PAO1 *P. aeruginosa clinical* isolates.	Reduced biofilm, pyocyanin, hemolysin, elastase, proteases, rhamno lipid production. Reduced swarming, swimming, and twitching motility	Zingerone (phytochemical from ginger)	Not provided in this study	[230]
Zeaxanthin	Anti-QS Anti-biofilm	*P. aeruginosa* PAO1	Reduced biofilm formation. Downregulated *rhlA* and *lasB* expression	Purified from the green alga *Chlorella ellipsoidea*	53.5 µM	[231]
Solonamides analogs	Anti-QS Anti-toxin	*S. aureus*	Reduced RNAIII and hla expression. Marginally enhanced biofilm formation	Cyclodepsipeptide isolated from the marine bacterium *Photobacterium halotolerans*	5 µg/mL	[232]
Flavonoids	Anti-QS	*P. aeruginosa* PA14	Reduced pyocyanin production and swarming motility. rhlA transcription inhibition	Naturally produced plant metabolites	Quercetin (100 µM): LasR activity ≈ 20–25% of control.	[233]
AIP analogs	Anti-QS Anti-biofilm	*S. epidermidis* RP62A	Reduced biofilm formation (using non-native agonist of *AgrC*-type I)	Analogs were created via solid-phase peptide synthesis	AIP-I (native)-99 nM AIP-I D1AS6A-18 nM	[234]
Coumarin	Anti-QS Anti-biofilm	*P. aeruginosa* PAO1 and clinical isolates	1. Reduced biofilm production, 2. Down-regulation of *lasI*, *rhlI*, *rhlR*, *pqsB*, *pqsC*, *pqsH*, *ambBCDE*, 3. Reduced protease and pyocyanin production 4. Reduced expression of T3SS secretion system-associated genes	Coumarin–hydroxamic acid conjugate	3.6 µM	[235]
Fluoro-substituted Isothiocyanates	Anti-QS	*P. aeruginosa*	1. Reduced pyocyanin production. 2. Reduced swarming motility. 3. Attenuated in vivo virulence of *P. aeruginosa* PAO1-UW toward *C.elegans*; 4. Attenuated *P. aeruginosa* P14 virulence in an *ex vivo* human skin burn wound model	Synthetic isothiocyanate- and maleimide-containing HSL-like small molecules	3.6 ± 1.9 µM	[236]
Pyridoxal lactohydrazone	Anti-QS	*P. aeruginosa* PAO1	Reduced swarming and twitching motility	Condensation of pyridoxal (vitamin B6) with lactic acid hydrazide	Sub-MIC concentrations (8 µg/mL and 32 µg/mL)	[237]
Glyceryl trinitrate	Anti-QS Anti-biofilm	*P. aeruginosa* PAO1 and clinical isolates	Reduced biofilm, pyocyanin and proteases production	Antivirulence compound	0.25 mg/mL (¼ MIC; sub-inhibitory)	[238]
Terrein	Anti-QS Anti-biofilm	*P. aeruginosa* PAO1	1. Reduced elastase, pyocyanin, rhamnolipid, and biofilm production; 2. Attenuated in vivo virulence of *P. aeruginosa* PAO1 toward *C. elegans* and mice	Secondary metabolite isolated from the fungus *Aspergillus terreus*	No half-maximal inhibitory concentration calculated	[239]
1,5-dihydropyrrol-2-ones analogs	Anti-QS	*E. coli* JB357 gfp reporter strain	QS inhibition	Thioether-linked dihydropyrrol-2-one analogs	32 µM	[240]
Parthenolide	Anti-QS Anti-biofilm	*P. aeruginosa* PAO1	Reduced pyocyanin, proteases, and biofilm production	Natural phytochemical	No	[241]
Diketopiperazine	Anti-QS Anti-biofilm	*B. cenocepacia*	1. Reduced biofilm formation. 2. Reduced protease and siderophore production	Synthetic diketopiperazine analogs	7.2 ± 0.2 µM	[242]
Lysionotin	Anti-toxin Anti-QS	*S. aureus*	1. Downregulate *fhla* and *gr* expression. 2. Reducedα-hemolysin production	Natural flavonoid	No	[243]
Methyl Eugenol-A	Interferes with AHL-regulated functions	*C. violaceum*	Inhibits EPS production, biofilm formation, and flagellar movement	Natural phenolic found in clove oil	0.2 mg/mL	[244]
Curcumin/10-undecenoic acid	Inhibit *LuxS/AI-2* and *LasI/LasR* QS systems	*P. aeruginosa*, *B. subtilis*	Counteracts bacterial pathogenicity and virulence	Natural polyphenolic compound	MIC values: 62.5 μg/mL	[245]
Ajoene	Downregulates QS genes	*P. aeruginosa*	Inhibits virulence factors and biofilm formation	Bioassay-guided fractionation of garlic oil	15 µM	[246]
Tanreqing (TRQ) formula	Inhibits upstream QS regulators (e.g., *GacS/GacA*, *PprA/PprB*)	*P. aeruginosa*	Inhibits virulence factors	Lab-prepared extract combining all five herbs (Huang Qin, Jin Yin Hua, Lian Qiao, Xiong Dan, Shan Yang Jiao)	Results are shown across TRQ dilutions (1/4, 1/8, 1/16 TRQ), not as IC_50_ values	[247]
Paecilomycone	Inhibits *PQS* and *HHQ* synthesis in *PQS* system	*P. aeruginosa*	Inhibits virulence factors	natural compound originally isolated from a fungal source-specifically *Paecilomyces* species	96.5 µM	[248]
Synthetic quorum-sensing inhibitors	Mimicking or blocking natural quorum-sensing signals to disrupt communication pathways.	*Ralstonia solanacearum* species complex	Impaired coordination of virulence factor expression, leading to diminished disease symptoms in host plants.	N-sulfonyl homoserine lactone analogs	1.66-4.91 µM	[249]
Ralfuranones	Interference with quorum-sensing signaling molecules, affecting the regulation of virulence factors.	*Ralstonia solanacearum* species complex	Decreased expression of virulence genes, resulting in reduced pathogenicity.	Derived from the 3-hydroxymethyl-2-methyl-4(1H)-quinolone	23.7 µM	[250]
DSF analogs	Compete with natural DSF molecules, disrupting normal QS signaling.	*S. maltophilia*, *P. aeruginosa*	Altered biofilm dynamics and reduced antibiotic resistance, diminishing virulence.	Natural DSF comes from *Xanthomonas*, *Burkholderia*, *Stenotrophomonas*, etc. via biosynthesis.	0.5 µM	[222]
Enzymatic degradation (e.g., DSF hydrolases)	Degrade DSF molecules, lowering their concentration and interfering with QS.	*E. carotovora* *P. aeruginosa*	Broad-spectrum reduction in QS-mediated virulence traits, including toxin production and biofilm formation.	*Stenotrophomonas maltophilia*	Not reported	[251]
Essential oils (e.g., clove, cinnamon, oregano)	Disrupt AHL-mediated signaling pathways, inhibiting signal synthesis and reception.	*P. fluorescens*, *S. putrefaciens*	Reduction in biofilm formation and extracellular enzyme production, leading to decreased spoilage activity.	Eugenol, β-caryophyllene	MIC: 0.2 mg/mL	[252]
Phenolic compounds (e.g., vanillin, eugenol)	Interfere with QS signal molecules and inhibit biofilm formation.	*A. hydrophila*, *L. monocytogenes*	Suppression of biofilm development and virulence factor expression, enhancing food safety.	Vanilla bean Clove, cinnamon, basil	MIC: 250 µg/mL MIC: 700 mg/L	[253]
Plant extracts (e.g., garlic, ginger)	Contain compounds that mimic or degrade QS signals, disrupting communication.	*E. coli*, *S. enterica*	Inhibition of QS-regulated behaviors, including toxin production and motility, reducing pathogenicity.	Allicin, ajoene and S-allyl cysteine	MIC: 0.325-0.625 mg/mL	[254]
Lactic acid bacteria metabolites	Produce bacteriocins and acids that interfere with QS systems.	*S. thermophiles*, *L. bulgaricus*	Decrease in QS activity, leading to reduced biofilm formation and spoilage potential.	Lactic acid, formic acid, folic acid, pyruvic acid, glutathione	Not reported	[255]
AHL analogs	Compete with natural AHLs for receptor binding, disrupting QS signaling.	*P. aeruginosa*, *E. coli*	Inhibition of biofilm formation and suppression of virulence gene expression.	Natural phenolic compound	100 µM	[10]
Furanones	Interfere with QS signal reception and stability.	*Vibrio* spp., *P. aeruginosa*	Reduction in biofilm development and attenuation of virulence factors.	Synthetic brominated furanone derivative	50 µM	[256]
Plant-derived compounds (e.g., garlic extract, cinnamaldehyde)	Inhibit QS signal synthesis and reception pathways.	*E. coli*, *S. enterica*	Decreased biofilm formation and prevention of VBNC state induction.	*Allium sativum*	0.325–0.625 mg/mL	[257]
Enzymatic degraders (e.g., AHL lactonases, acylases)	Degrade QS signaling molecules, disrupting communication.	*E. coli*, *S. enterica*	Impaired biofilm maturation and reduced virulence expression.	AHL lactonase (AiiA) Bacillus sps	Not reported	[258]
Adenosine	Modulates QS pathways, potentially interfering with signal molecule synthesis or reception.	*Cylospora*	May reduce expression of virulence factors and biofilm formation, enhancing host immune response.	Endogenous nucleoside; found in all living cells	Not reported	[259]
AI-2, DSF family signals	Biofilm formation, motility, and antibiotic resistance	*S. aureus*, *E. coli*	Decrease in QS-regulated behaviors, including toxin production and biofilm formation.	Universally produced by many Gram-negative and Gram-positive bacteria	Not applicable	[80]
Probiotic-produced Autoinducers	Interfere with pathogenic QS signals by competitive binding or signal degradation, disrupting bacterial communication.	*E. coli*, *S. enterica*, *C. difficile*	Reduced biofilm formation and downregulation of virulence gene expression.	*Lactobacillus*, *Bifidobacterium*, *E. coli* Nissle 1917	Not reported	[260]
Probiotic biofilm formation	Colonizes intestinal mucosa to outcompete pathogens for adhesion, indirectly suppressing QS-mediated virulence.	*E. faecalis*, *S. aureus*	Limits pathogen colonization and strengthens mucosal barrier function	*Enterococcus faecalis*, *Lactobacillus* spp., *Bifidobacterium* spp.	Not reported	[261]
Plant-derived QS inhibitors (e.g., flavonoids, phenolics)	Block QS signal synthesis or reception, degrade signaling molecules.	*P. aeruginosa*, *Vibrio* spp.	Inhibits biofilm development and virulence factor production.	Various plant sources	50 µM	[262]
Immune modulation by probiotics	Stimulates host immune responses, producing cytokines that counteract infection and inflammation.	*S. enterica*, *C. difficile*	Supports intestinal barrier integrity and suppresses pathogen-induced inflammation.	Citrus fruits, green tea, various plants	50–100 µM	[263]
Synthetic QS inhibitors	Engineered molecules or circuits designed to interfere with QS signaling pathways.	*P. aeruginosa*	Disruption of QS-mediated behaviors such as biofilm formation, virulence factor production, and antibiotic resistance.	Engineered small molecules or genetic circuits designed via synthetic biology	10–100 µM for synthetic analogs	[264]
Probiotic interference	Probiotic strains engineered to produce QS inhibitors or compete with pathogens for QS signals.	*E. coli*, *S. enterica*, *C. difficile*	Reduction in pathogen colonization and virulence through competitive inhibition and signal interference.	Engineered or natural probiotic strains (*Lactobacillus*, *Bifidobacterium*, *E. coli* Nissle 1917)	Rarely specified	[265]
Phage-mediated QS disruption	Bacteriophages engineered to degrade QS signals or produce QS inhibitors.	*P. aeruginosa*	Decreased biofilm formation and virulence factor expression through targeted QS disruption.	Engineered bacteriophages designed to degrade QS signals or produce QS inhibitors	Not reported	[266]
Sulfonamide-based DSF analogs	Interfere with DSF receptor binding, disrupting DSF-mediated QS.	*S. maltophilia*, *B. cepacia*	Reduced biofilm formation and virulence factor production.	Chemically synthesized small molecules mimicking DSF structure	5–50 µM	[267]
Retinoic acid (RA)	Modulates DSF-mediated QS, affecting biofilm formation and motility.	*S. maltophilia*	Decreased biofilm formation and motility, enhancing antimicrobial efficacy.	Naturally occurring metabolite of Vitamin A	50 µM	[268]
Endogenous DSF analog	Mimics DSF, disrupting DSF-mediated QS.	*S. maltophilia*	Inhibition of biofilm formation and motility.	Naturally produced by bacteria or chemically synthesized to mimic DSF	Not reported	[269]
Coumarin–chalcone conjugate	Anti-QS Anti-biofilm	*P. eruginosa*	Decreased biofilm formation and virulence factor expression through targeted QS disruption.	Chemically synthesized conjugate combining coumarin and chalcone scaffolds	20–50 µM	[270]
QteE	Sequesters *LasR*, increasing the threshold for QS activation.	*P. aeruginosa*	Delayed expression of QS controlled genes, reducing early activation of virulence factors.	Protein from *Pseudomonas aeruginosa*	Not reported	[271]
QscR	Binds to *LasR* and RhlR, preventing premature activation of QS.	*P. aeruginosa*	Suppressed expression of QS controlled genes, mitigating early virulence factor production.	Transcriptional regulator from *Pseudomonas aeruginosa*	56–62 nM	[221]
QslA	Inhibits *LasR* and *RhlR* activity, modulating QS response.	*P. aeruginosa*	Reduced biofilm formation and virulence factor production.	Transcriptional regulator from *Pseudomonas aeruginosa*	Not reported	[272]
Tyramine	Quorum-sensing inhibition via interference with the *CepIR* system	*B. cenocepacia*.	Decreased biofilm formation, reduced motility, attenuated virulence in *Caenorhabditis elegans* model.	Naturally occurring biogenic amine	Not reported	[273]
QStatin	Specifically targets LuxO, a central response regulator in the quorum-sensing pathway of *Vibrio* species. QStatin binds selectively to phosphorylated LuxO, blocking its ability to activate small regulatory RNAs (sRNAs), which are essential for downstream quorum-sensing responses.	*V. cholerae* *V. harveyi*	Disrupts quorum-sensing-controlled gene expression Reduces virulence-related phenotypes Selective: QStatin does not inhibit general bacterial growth or non-target bacteria significantly	Small-molecule inhibitor designed to selectively target LuxO	2–5 µM	[274]
Furanones	Competitive inhibition of AHL receptor proteins (LuxR-type regulators)	*P. aeruginosa*, *E. coli*, *V.* spp.	Reduces biofilm formation, motility, and production of virulence factors (e.g., toxins)	Synthetic halogenated furanone	3–9 µM	[275]
Baicalein	Downregulates quorum-sensing-regulated gene expression	*P. aeruginosa*, *S. aureus*	Inhibits biofilm formation and reduces toxin production	Flavonoid from *Scutellaria baicalensis* and other plants	Not specified	[276]
Quercetin	Interferes with AI-2 signaling pathways	*E. coli*, *S.* spp.	Inhibits quorum-sensing gene expression, reduces swarming, and suppresses virulence	Naturally occurring flavonoid	281 µM for biofilm inhibition	[227]
Vanillin	Blocks AHL signal molecule activity	*C. violaceum*, *P. aeruginosa*	Inhibits violacein pigment production and biofilm formation	Natural compound from vanilla beans	IC_50_ = 0.81 mM; MIC = 16 mM	[277]
Furanone C-30	Inhibits LasR receptor in the *las* quorum-sensing system	*P. aeruginosa*	Reduces biofilm formation, elastase activity, and enhances tobramycin susceptibility (initially)	Synthetic brominated furanone	IC_50_: 3–9 µM	[278]
QS Inhibitor-loaded biomimetic nanoparticles	Inhibits AI-2 quorum-sensing system, disrupts cell communication + induces ferroptosis-like bacterial death via lipid peroxidation	Methicillin-Resistant *S. aureus* (MRSA)	Suppresses virulence gene expression, reduces biofilm formation, and promotes bacterial death in infected lung tissue	Natural compounds or synthetic analogs	2–5 µM	[279]
L-mimosine	Inhibits quorum-sensing-regulatory small RNAs (sRNAs) specifically, qrr sRNAs in *Vibrio* species	*V. harveyi* *E. coli* *P. aeruginosa*	Reduces bioluminescence (in *Vibrio*), biofilm formation, motility, and toxin production	*Mimosa pudica* and *Leucaena* species	Not well reported	[280]
2-(4-(acridin 9yl amino)phenyl)isoindoline-1,3-dione derivatives	Inhibits the Pseudomonas Quinolone Signal (PQS) quorum-sensing system	*P. aeruginosa*	Reduces virulence-related phenotypes	Acridine-based isoindoline derivatives	Not well reported	[281]

### 4.4. Phage Therapy and Bacteriophage-Derived Enzymes: Precision Tools Against Biofilms

Bacteriophage therapy, once regarded as a last option, is currently experiencing a resurgence, driven by advancements in synthetic biology and precise engineering. Virus-based therapies that selectively infect and eliminate bacteria are demonstrating exceptional efficacy in addressing biofilms, particularly those resistant to antibiotics (Figure 4c) [282,283]. Contemporary advancements have converted phages into programmable precision instruments. CRISPR-Cas-modified bacteriophages may now selectively cleave antibiotic-resistance genes within biofilm communities. This method eradicates resistant bacteria while safeguarding commensal species, presenting a significant benefit over broad-spectrum antibiotics [284]. Through the engineering of phage capsids and tail fibers, researchers have broadened host ranges and improved matrix penetration, facilitating the efficient eradication of polymicrobial biofilms [285]. The generation of lysis-deficient phages is one of the most significant advancements. In contrast to conventional phages that lyse host cells, these engineered viruses eliminate bacteria by inducing intracellular toxin production, hence preventing the discharge of inflammatory debris. In prosthetic-joint infection models, these phages attained 3 to 5 log decreases in biofilm load without inducing immune over activation [286]. This invention is particularly pertinent in delicate locations where inflammation must be minimized. In addition to phage therapy, phage-derived enzymes are becoming acknowledged as effective antibiofilm agents. These encompass polysaccharide depolymerases that particularly degrade essential constituents of the biofilm matrix, including alginate in *P. aeruginosa*, PNAG in *S. aureus*, and cellulose in *E. coli* [287]. Fusion enzymes, which integrate matrix-degrading activity with antimicrobial peptides, are even more potent as they concurrently dismantle structure and eliminate embedded cells [288]. These enzymes are currently being encapsulated in nanoparticles for prolonged release at difficult-to-access infection locations such as implants.

The most promising method may be phage antibiotic synergy (PAS). This method utilizes phages to disrupt biofilms and inhibit resistance mechanisms, while sub-lethal antibiotics promote phage replication by inducing stress in the bacteria. In clinical research, PAS has elevated cure rates in *K. pneumoniae* catheter biofilms from 31% with antibiotics alone to 78% when administered in conjunction [289]. AI-driven algorithms are currently being created to identify appropriate phage antibiotic combinations based on bacterial genetics and behaviors, facilitating tailored PAS regimens [290]. Phage treatment is fundamentally transforming the principles of antimicrobial intervention. Bacteriophages offer a physiologically selective, self-replicating, and adaptable solution to the biofilm issue, utilizing tailored lytic precision, matrix-targeting enzymes, or synergistic combinations [291]. With the increasing clinical interest, these developments are set to become essential elements of next-generation infectious disease management [292].

### 4.5. Host-Directed Therapies and Immune Modulation: Rewiring Defenses Against Biofilms

Conventional biofilm medicines directly target bacteria; however, a powerful alternative is emerging that enhances the human immune system to fight more effectively. Biofilms are known for forming an immunosuppressive environment that inhibits phagocytosis, modifies cytokine profiles, and shields infectious pathogens from being eliminated. Host-directed treatments (HDTs) seek to counteract immunological infectiveness [293] and rescue the body’s innate immune defense mechanisms [294]. Recent findings in immunometabolism have identified critical sites for immunological reprogramming. In the context of chronic wounds, bacteria biofilm-secreted proteins prompt macrophages to assume an M2 (anti-inflammatory) phenotype, thereby suppressing antimicrobial responses and hindering the migration of other immune cells to control the infection [295]. Nonetheless, low-dose everolimus, a mTOR pathway inhibitor, has demonstrated the capacity to revert macrophages to the pro-inflammatory M1 phenotype [296], thus enhancing their ability to produce reactive oxygen species and eradicate biofilm infections by as much as fivefold [297]. Cytokine treatments, including IL-12 in conjunction with GM-CSF, have reinstated Th1 immune responses, augmented neutrophil extracellular trap (NET) production, and significantly diminished *S. aureus* biofilms in implant-associated infections (Figure 4d) [298]. Apart from small compounds, an emerging array of targeted biologics and modified immune cells is broadening the potential for host regulation. Monoclonal antibody F598 inhibits *P. aeruginosa’s LecA*-mediated immune suppression, demonstrating the first therapeutic efficacy in ventilator-associated pneumonia. Simultaneously, nanovesicle-based decoys are being engineered to bind and neutralize immunosuppressive substances released by biofilms, thereby restoring neutrophil activity in cystic fibrosis airway models. CRISPR-edited neutrophils, which have been modified to exhibit enhanced surface markers such as CD11b, demonstrate improved motility and penetration into biofilm matrices by up to 40% [299].

The development of vaccines is also garnering interest. Researchers are developing next-generation vaccines that provoke protective, biofilm-specific immune responses by targeting matrix components and stress-induced biofilm proteins. These may be especially beneficial for high-risk individuals experiencing repeated device-related or respiratory infections [22]. Precision medicine enhances the efficacy of host-directed interventions. Patient-derived three-dimensional organoids and autologous immune cells now provide individualized ex vivo evaluation of immunomodulators [295]. Machine learning methods are utilized to simulate immune responses and inform cytokine therapy choices based on individual immunoprofiles. In a clinical investigation, ex vivo primed macrophage therapy resulted in a 75% increase in biofilm clearance in diabetic foot infections vs. usual treatment [300]. Collectively, these host-centric developments are transforming the paradigm of biofilm therapy from antibiotic eradication to immunological rescue. HDTs provide sustainable, microbiome-conserving, and highly individualized therapies by addressing the fundamental immune malfunction that enables the persistence of biofilms [147]. As these techniques advance, they are set to become essential adjuncts to direct-acting antimicrobials in the management of complex, chronic illnesses.

## 5. Translational Challenges and Future Directions in Biofilm Therapeutics

The transition from laboratory research to clinical use in biofilm therapies presents not only a technical obstacle but also regulatory, ethical, and societal challenges. Next-generation therapies, such as modified phages, CRISPR-based antimicrobials, and biofilm-responsive nanotechnologies, exhibit significant potential; yet their trajectory toward clinical implementation is intricate and disjointed [133]. This section delineates the principal problems and forthcoming priorities influencing the domain of biofilm medicine.

### 5.1. Regulatory Bottlenecks and Standardization Gaps

Existing regulatory frameworks mostly focus on traditional small-molecule antibiotics, resulting in a gap in the assessment of intricate biologics and synthetic structures. Engineered phages, CRISPR nanocarriers, and probiotic therapies frequently lack definitive approval processes owing to their multifunctional and dynamic characteristics. Although attempts like the expedited approval of CTX-2024, a CRISPR-enhanced phage treatment for cystic fibrosis biofilms, indicate advancement, these regulatory frameworks are used inconsistently across different areas and therapeutic categories [301]. There is an urgent requirement for standardized global protocols that can address the complexities of biologics targeting biofilms. This entails the delineation of biofilm-specific clinical objectives (e.g., biomass reduction, matrix degradation, recurrence suppression) and the formulation of rigorous protocols for post-market surveillance, especially for treatments exhibiting environmental persistence [302].

### 5.2. Ethical and Biosafety Considerations

The introduction of genetically modified organisms presents numerous biosafety and ethical challenges. Issues related to horizontal gene transfer, off-target impacts, and unforeseen ecological repercussions necessitate the incorporation of fail-safe design elements, including kill-switch circuits, auxotrophic dependencies, and non-replicative vectors [303]. The WHO’s 2024 consensus framework has advanced by excluding heritable gene modifications and requiring temporary, self-limiting structures for clinical applications. In addition to biosafety, public perception and trust are essential. Surveys reveal more endorsement for engineered probiotics in therapeutic settings compared to environmental uses, highlighting the necessity for clear scientific communication and inclusive stakeholder involvement [304]. Ethical success will rely on responsible innovation and the establishment of societal consensus via proactive discussion.

### 5.3. Microbiome-Preserving Strategies and Precision Design

A significant progression in next-generation therapeutics is the shift towards microbiome-sparing approaches. In contrast to broad-spectrum antibiotics, these approaches specifically target pathogenic biofilms while safeguarding commensal microbial communities. For instance, *PslG depolymerase* specifically targets the matrix polysaccharides of *P. aeruginosa* with little collateral effects [305], whereas tailored probiotics and metabolite-guided therapy have shown an over 85% decrease in pathogens without altering microbiota composition [306]. The integration of artificial intelligence, nanotechnology, and synthetic biology is facilitating enhanced precision. Examples comprise machine learning-optimized antimicrobial peptides designed for matrix penetration, and CRISPR nanocarriers that administer both gene editing and medicinal payloads with spatial precision [301].

### 5.4. Towards Real-Time, Autonomous Therapies

Looking forward, biofilm management may be revolutionized by autonomous, closed-loop therapeutic platforms. These systems integrate biosensors, responsive delivery mechanisms, and adaptive algorithms to continuously monitor and modulate therapy in real time. Early models show promise in eradicating biofilms through on-demand activation triggered by environmental signals [307]. However, significant technical, safety, and validation hurdles remain before clinical integration becomes feasible.

### 5.5. Pathways to Clinical Translation

Bridging the translational gap in biofilm therapeutics demands a unified, interdisciplinary approach. Developing standardized preclinical models that replicate human tissues, immune responses, and microbiota is critical for predicting clinical efficacy [308]. Collaboration among researchers, clinicians, regulators, and ethicists will ensure that novel interventions are both effective and ethically sound. Robust evaluation metrics must be refined to assess not only bacterial clearance but also long-term safety, resistance suppression, and tissue recovery [309]. Cross-sector alignment is essential to streamline regulatory pathways for emerging biologics. Public–private partnerships can facilitate resource-sharing and accelerate innovation. Equally, iterative clinical feedback loops must inform early-stage design. Transparency in biosafety practices will build public trust. Investment in education and policy will support broader adoption. Ultimately, success hinges on translating precision science into safe, scalable, and societally accepted biofilm solutions.

## 6. Conclusions

Biofilms present a major challenge in modern microbiology, tightly linked to antibiotic resistance, persistent infections, and therapeutic failure. Our thorough investigation indicates that conventional antibiotic strategies are inadequate to infiltrate the protective matrix, metabolic dormancy, and collaborative behaviors inherent in biofilm ecosystems. The integration of synthetic biology, nanomedicine, artificial intelligence, and host-directed therapeutics has initiated a novel era of precision antibiofilm methods.

We emphasize the future interventions that are driven by machine learning-based drug discovery, CRISPR-modified phages, and responsive nanoparticles, all of which can offer support in providing tailored interventions that affect biofilm architecture, communication, and metabolic control. These methods are not only more efficacious but are also engineered to safeguard the microbiome, thereby reducing off-target harm and the emergence of resistance. The integration of host-directed therapies, including cytokine modulation and modified immune cells, signifies a transformative shift towards enhancing the host immune response, thus mitigating the immunosuppressive microenvironments established by biofilms. Progress relies not just on technological advancements but also on system-level integration via standardized preclinical models, ethical biosafety protocols, and iterative clinical feedback. The creation of real-time, autonomous treatment systems with environmental sensing and adaptive response capabilities will expedite the clinical application of these sophisticated techniques. In conclusion, eliminating the biofilm stronghold necessitates a multifaceted strategy that integrates microbial precision, host resilience, and regulatory foresight. As we progress towards a post-antibiotic era, the integration of cross-disciplinary innovation presents a viable approach to sustainable, scalable, and socially accepted biofilm treatments.

## Figures and Tables

**Figure 1 antibiotics-14-00865-f001:**
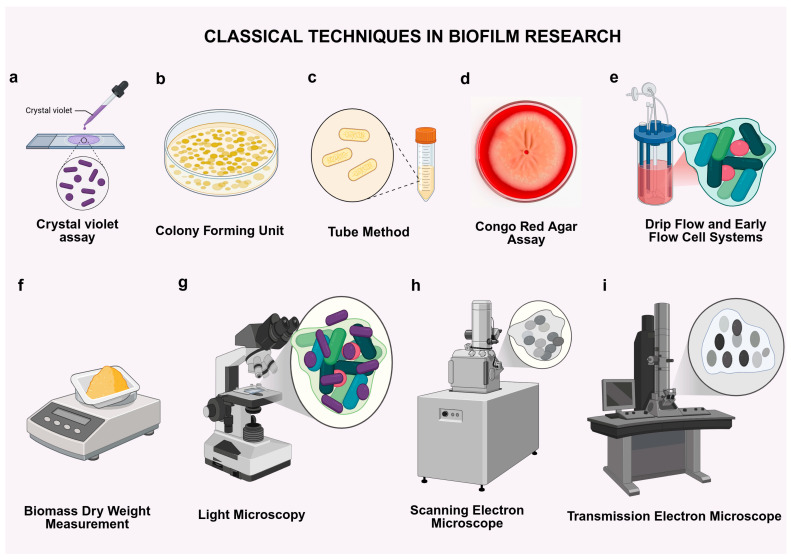
Classical techniques in biofilm research: Created in Biorender. Bhasme, P. (2025). https://BioRender.com/b67h2c1.

**Figure 2 antibiotics-14-00865-f002:**
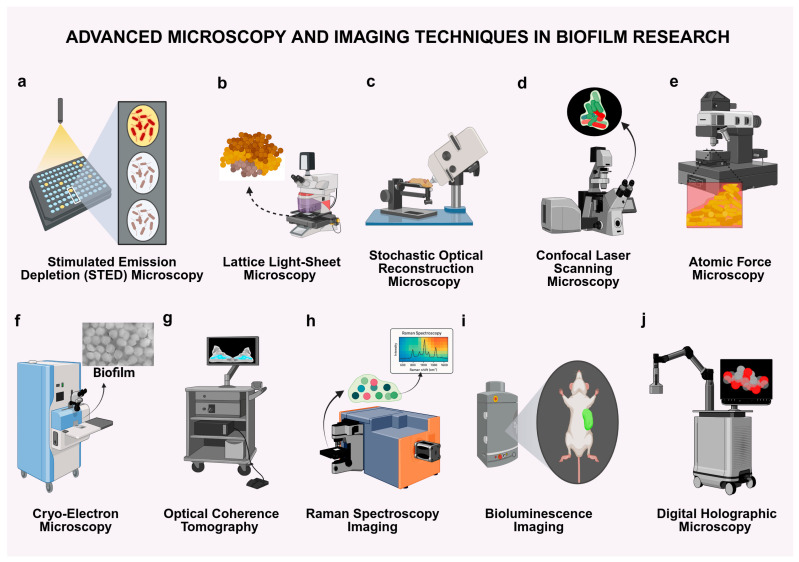
Advanced microscopy and imaging techniques in biofilm research: Created in Biorender. Bhasme, P. (2025). https://BioRender.com/b67h2c1.

**Figure 3 antibiotics-14-00865-f003:**
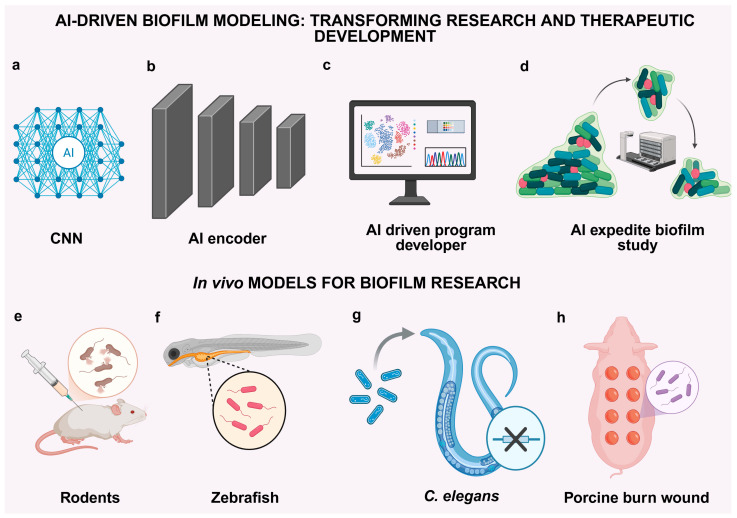
In vivo and AI technology in biofilm research: Created in Biorender. Bhasme, P. (2025). https://BioRender.com/b67h2c1.

**Figure 4 antibiotics-14-00865-f004:**
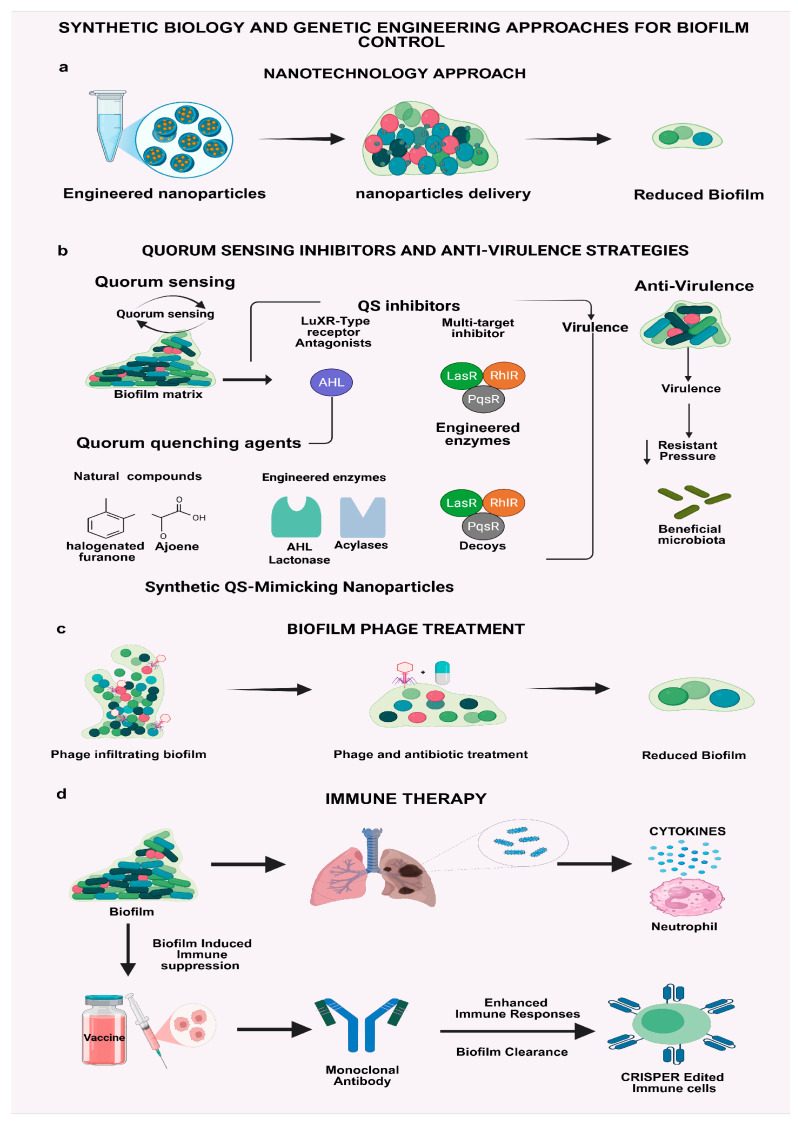
Synthetic biology and genetic engineering approaches for biofilm control: Created in Biorender. Bhasme, P. (2025). https://BioRender.com/b67h2c1.

## Data Availability

The dataset supporting the findings of this study is included within the manuscript and its referenced sources, ensuring comprehensive access to the relevant data for further examination and analysis.

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
