# Peer review of "Biofilms Exposed: Innovative Imaging and Therapeutic Platforms for Persistent Infections"

_antibiotics, 2025, doi:10.3390/antibiotics14090865_

Round 1
Reviewer 1 Report
Comments and Suggestions for Authors
this is an interesting and comprehensive review of what makes the study of biofilms so important. It clearly describes the techniques that are in use today for such studies from classical work to technological advanced methodologies and AI platforms. It further provides novel therapeutic methodologies for reducing and eliminating pathogenic biofilms. The authors then go further and suggest treatments based on quorum sensing inhibition or enhancement and adds a section on phage therapeutics They furthermore recognize the limitation to using some of these methodologies
In all I find this work to be excellent well written and ready for publication as is.
Author Response
Thank you for accepting our paper.
Reviewer 2 Report
Comments and Suggestions for Authors
The authors have put together a comprehensive review of approaches that can be used to disrupt biofilms during the treatment of infectious diseases. It is quite thorough with over 300 references. There are several adjustments that can be made that will benefit readers of the review.
First, the tables (which have a lot of valuable information) do not appear to be listed in the text and should be in appropriate places. For example, Table 1 which is a 'Summary of Advanced Imaging Techniques in Biofilm Research' should be referenced in section 3.1 Advanced Microscopy and Imaging Techniques in Biofilm Research, along with the referenced Figure 2. Similarly, Table 2. Therapueutic Nanomaterials Targeting Biofilm Forming Bacteria probably should be referred to in section 4.2 Nanotechnology Approaches for Biofilm Control, and Table 3. Anti-Quorum Sensing Materials to Treat Biofilm Producing Bacteria should probably be referred to in section 4.3 Quorum Sensing Inhibitors and Anti-Virulence Strategies: Disrupting Bacterial Communication. They were not referred to in the manuscript that was reviewed.
Second, Figure 2a-d looks like its a flow-chart proceeding from CNN, to AI encoder, to AI driven program developer, to Biofilm developer. However, the text describing this portion of the figure does not really connect a-d. It is possible that the authors did not intend for Figure 2 a-d to be a flow-chart, and if not, the arrows should be removed and each section stand alone. If, on the other hand, they did want it to be a flow-chart, then the corresponding text should be modified to show the connections.
Third, Abbreviations. Listing the abbreviations is a good idea because it gives readers a place to go to in order to reconnect with ahat the abbreviations stand for. One suggestion would be to change the listing from the order of appearance in the manuscript to an alphabetical listing. readers may find it easier to locate the abbreviation that they are looking for. Similarly, also list the bacteria alphabetically.
Minor points:
Line 68, change 'per sister' to 'persister'
Section 2.2, lines 116-117, unfortunately, the presence of a CFU growing on a selective medium does not really signify that the bacterium was viable in the location it was retrieved from.
Line 326, change to 'C. elegans, a nematode, has also...'
Line 327, perhaps change the writing to 'which can explain the interaction between bacteria and invertebrates'
Lines 327-328, 'Yersina pseudotuberculosis' should be italicized.
Section 4.3. It might be valuable to the readers to briefly remind (or inform) them what the various genes that are referenced actually do (i.e., LasR, Las, Rhl and PQS and any other genes listed in the text).
Author Response
Comment :
First, the tables (which have a lot of valuable information) do not appear to be listed in the text and should be in appropriate places. For example, Table 1 which is a 'Summary of Advanced Imaging Techniques in Biofilm Research' should be referenced in section 3.1 Advanced Microscopy and Imaging Techniques in Biofilm Research, along with the referenced Figure 2. Similarly, Table 2. Therapueutic Nanomaterials Targeting Biofilm Forming Bacteria probably should be referred to in section 4.2 Nanotechnology Approaches for Biofilm Control, and Table 3. Anti-Quorum Sensing Materials to Treat Biofilm Producing Bacteria should probably be referred to in section 4.3 Quorum Sensing Inhibitors and Anti-Virulence Strategies: Disrupting Bacterial Communication. They were not referred to in the manuscript that was reviewed.
Response :
Thank you for this observation. We have now incorporated in-text citations for all the tables at their appropriate locations. Specifically, Table 1 is referenced in Section 3.1 alongside Figure 2; Table 2 is cited in Section 4.2; and Table 3 is referenced in Section 4.3. these are all highlighted with green
Comment :
Second, Figure 2a-d looks like its a flow-chart proceeding from CNN, to AI encoder, to AI driven program developer, to Biofilm developer. However, the text describing this portion of the figure does not really connect a-d. It is possible that the authors did not intend for Figure 2 a-d to be a flow-chart, and if not, the arrows should be removed and each section stand alone. If, on the other hand, they did want it to be a flow-chart, then the corresponding text should be modified to show the connections
Response :
We appreciate this comment. Figure 2a-d was not intended as a linear flowchart. We have now revised the figure to remove the connecting arrows and updated the legend accordingly. Each panel is now presented as an independent visual representation.
Comment :
Third, Abbreviations. Listing the abbreviations is a good idea because it gives readers a
place to go to in order to reconnect with ahat the abbreviations stand for. One suggestion
would be to change the listing from the order of appearance in the manuscript to an
alphabetical listing. readers may find it easier to locate the abbreviation that they are
looking for. Similarly, also list the bacteria alphabetically.
Response:
We arranged all the abbreviation in alphabetical orders, thank you
Minor points:
We changed per sister to persister
Comment :
Section 2.2, lines 116-117, unfortunately, the presence of a CFU growing on a selective medium does not really signify that the bacterium was viable in the location it was retrieved from.
Response :
We revised the sentences and added information in line 132 to 136 and highlighted in green
Comment :
Line 326, change to 'C. elegans, a nematode, has also...'
Response :
We corrected C. elegans to italic and highlighted in green text in the draft and italicized the Yersina pseudotuberculosis'
Comment :
Line 327, perhaps change the writing to 'which can explain the interaction between bacteria and invertebrates'
Response: We corrected the text and provided information in line 354 and 355
Comment:
Section 4.3. It might be valuable to the readers to briefly remind (or inform) them what the
various genes that are referenced actually do (i.e., LasR, Las, Rhl and PQS and any other
genes listed in the text).
Response:
We incorporated brief information about LasR, Las, Rhl and PQS in line 457 to 460and highlighted in green
Reviewer 3 Report
Comments and Suggestions for Authors
Following are my comments to assist the editor in making a decision:
-
The author should provide statistical data regarding the utility of in vivo and ex vivo models that replicate clinical biofilms. Specifically, how many studies have reported on this to date?
-
Similarly, statistical information should be presented regarding the number of studies published to date on recent advances in biofilm-targeted therapies—such as CRISPR-Cas-modified bacteriophages, quorum sensing inhibitors, enzyme-functionalized nanocarriers, and intelligent drug delivery systems responsive to biofilm-specific cues.
-
I recommend that the author clearly outline the objectives of the review in a pointwise format at the end of the Introduction section.
-
Figures 1 to 4 appear to contain multiple illustrations created using BioRender. However, the author has not acknowledged the source or included a copyright statement. This should be corrected.
-
In Section 3.1, titled "Advanced Microscopy and Imaging Techniques in Biofilm Research", merely including an image is insufficient. The author should elaborate on the technological progress and emerging developments in these instruments, particularly in the context of biofilm research and control.
-
Table 1 lacks citations for the included information. References must be provided.
-
In Table 2, the author should specify the raw materials used for the nanomaterials, the active concentrations applied, and the minimum biofilm inhibitory concentrations (MBIC) reported for each material.
-
For Table 3, the sources of the compounds listed and their active inhibitory concentrations against biofilms should be provided.
-
Instead of using general images of organs and instruments, the author should include mechanistic figures or schematic diagrams that illustrate how the discussed therapies or technologies function against biofilms.
The author must seek professional language editing.
Author Response
- The author should provide statistical data regarding the utility of in vivo and ex vivo
models that replicate clinical biofilms. Specifically, how many studies have reported
on this to date?
Response:
Thank you for the insightful suggestion. We have now added information in Section 3.3 summarizing the number of published studies utilizing in vivo and ex vivo biofilm models. Information is highlighted in red
- Similarly, statistical information should be presented regarding the number of
studies published to date on recent advances in biofilm-targeted therapies—such as
CRISPR-Cas-modified bacteriophages, quorum sensing inhibitors, enzyme functionalized nanocarriers, and intelligent drug delivery systems responsive to biofilm-specific cues.
Response :
Thank you for the insightful suggestion. We have now added information in Section 4.1 summarizing the number of published studies utilizing in vivo and ex vivo biofilm models. Information is highlighted in red
- I recommend that the author clearly outline the objectives of the review in a
pointwise format at the end of the Introduction section.
Response :
Thank you for the suggestion. We have now added outlines of the objectives of the review in a pointwise format at the end of the Introduction section and highlighted in red
- Figures 1 to 4 appear to contain multiple illustrations created using BioRender.
However, the author has not acknowledged the source or included a copyright
statement. This should be corrected.
Response:
Thank you for pointing this out. We have now added the appropriate attribution for figures created with BioRender.com.
- In Section 3.1, titled "Advanced Microscopy and Imaging Techniques in Biofilm
Research", merely including an image is insufficient. The author should elaborate
on the technological progress and emerging developments in these instruments,
particularly in the context of biofilm research and control.
Response :
Thank you so much for your very thoughtful comments. Here we were trying to show the what are the microscope can be used for the study biofilm. Considering your comments now we modified the figure 2 Advanced Microscopy and Imaging Techniques in Biofilm Research
- Table 1 lacks citations for the included information. References must be provided.
Response:
We have now included proper citations in Table 1 for each imaging technique listed, referencing the original studies or review articles from which the data were derived.
- In Table 2, the author should specify the raw materials used for the nanomaterials,
the active concentrations applied, and the minimum biofilm inhibitory
concentrations (MBIC) reported for each material.
Response: AS per the suggestions Table 2 has been revised to include the raw material base of each nanomaterial, its active concentration, and where available, the reported minimum biofilm inhibitory concentration (MBIC) and highlighted in red
- For Table 3, the sources of the compounds listed and their active inhibitory
concentrations against biofilms should be provided.
Response: Thank you for the suggestion. We have now revised Table 3 to include both the natural or synthetic source of each anti-quorum sensing compound and their reported inhibitory concentrations against biofilm-forming pathogens and highlighted in red
- Instead of using general images of organs and instruments, the author should
include mechanistic figures or schematic diagrams that illustrate how the discussed
therapies or technologies function against biofilms.
Response: We appreciate this comment. We have replaced generic illustrations with schematic figures that represent the mechanism of action for each therapeutic approach, including QS inhibition, phage delivery, and nanoparticle-biofilm interactions.
Round 2
Reviewer 3 Report
Comments and Suggestions for Authors
accept